# Towards physically consistent data-driven weather forecasting: Integrating data assimilation with geometric deep learning in a case study with ERA5

Ashesh Chattopadhyay[1,2], Mustafa Mustafa[2], Pedram Hassanzadeh[1,3], Eviatar Bach[4,5], and Karthik Kashinath[2]

[1]Department of Mechanical Engineering, Rice University, Houston, TX, USA
[2]Lawrence Berkeley National Laboratory, Berkeley, CA, USA
[3]Department of Earth, Environmental and Planetary Sciences, Rice University, Houston, TX, USA
[4]Department of Atmospheric and Oceanic Science and Institute for Physical Science and Technology, University of Maryland, College Park, USA
[5]Geosciences Department and Laboratoire de Météorologie Dynamique (CNRS and IPSL), École Normale Supérieure and PSL University, Paris, France

**Correspondence:** Pedram Hassanzadeh (pedram@rice.edu)

**Abstract.** There is growing interest in data-driven weather prediction (DDWP), for example using convolutional networks such as U-NETs that are trained on data from models or reanalysis. Here, we propose 3 components to integrate with commonly used DDWP models in order to improve their physical consistency and forecast accuracy. These components are 1) a deep spatial transformer added to the latent space of the U-NETs to preserve a property called equivariance (a concept in geometric deep learning), which is related to correctly capturing rotations and scalings of features in spatio-temporal data, 2) a data-assimilation (DA) algorithm to ingest noisy observations and improve the initial conditions for next forecasts, and 3) a multi-time-step algorithm, which combines forecasts from DDWP models with different time steps through DA, improving the accuracy of forecasts at short intervals. To show the benefit/feasibility of each component, we use geopotential height at 500 hPa (Z500) from ERA5 reanalysis and examine the short-term forecast accuracy of specific setups of the DDWP framework. Results show that the equivariance-preserving networks (U-STNs) clearly outperform the U-NETs, for example improving the forecast skill by $45\%$. Using a sigma-point ensemble Kalman (SPEnKF) algorithm for DA and U-STN as the forward model, we show that stable, accurate DA cycles are achieved even with high observation noise. This DDWP+DA framework substantially benefits from large ($O(1000)$) ensembles that are inexpensively generated with the data-driven forward model in each DA cycle. The multi-time-step DDWP+DA framework also shows promises, e.g., it reduces the average error by factors of 2-3. These results show the benefits/feasibilities of these 3 components, which are flexible and can be used in a variety of DDWP setups. Furthermore, while here we focus on weather forecasting, the 3 components can be readily adopted for other parts of the Earth system, such as ocean and land, for which there is a rapid growth of data and need for forecast/assimilation.

# 1 Introduction

Motivated by improving weather and climate prediction, using machine learning (ML) for data-driven spatio-temporal fore-
casting of chaotic dynamical systems and turbulent flows has received substantial attention in recent years (e.g., Pathak et al.,
2018; Vlachas et al., 2018; Dueben and Bauer, 2018; Scher and Messori, 2018, 2019; Chattopadhyay et al., 2020b, c; Nadiga,
2020; Maulik et al., 2021). These data-driven weather prediction (DDWP) models leverage ML methods such as convolutional
neural networks (CNNs) and/or recurrent neural networks (RNNs) that are trained on state variables representing the history
of the spatio-temporal variability, and learn to predict the future states (we have briefly described some of the technical ML
terms in Table 1). In fact, a few studies have already shown promising results with DDWP models that are trained on variables
representing the large-scale circulation obtained from numerical models or reanalysis products (Scher, 2018; Chattopadhyay
et al., 2020a; Weyn et al., 2019, 2020; Rasp et al., 2020; Arcomano et al., 2020; Rasp and Thuerey, 2021, e.g.,). Chattopad-
hyay et al. (2020d) showed DDWP models trained on general circulation models' (GCM) outputs can be used to predict
extreme temperature events. Excellent reviews and opinion pieces on the state-of-the-art of DDWP can be found in Chantry
et al. (2021), Watson-Parris (2021), and Irrgang et al. (2021). Other applications of DDWP may include post-processing of
ensembles (Grönquist et al., 2021) and sub-seasonal to seasonal prediction (Scher and Messori, 2021; Weyn et al., 2021).

The increasing interest (Schultz et al., 2021; Balaji, 2021) in these DDWP models stems from the hope that they improve
weather forecasting because of one or both of the following reasons: 1) trained on reanalysis data and/or data from high-
resolution NWP models, these DDWP models may not suffer from some of the biases (or generally, model error) of physics-
based, operational numerical weather prediction (NWP) models, and 2) the low computational cost of these DDWP models
enables generating large ensembles for probabilistic forecasting (Weyn et al., 2020, 2021). Regarding (1), while DDWP models
trained on reanalysis data have skills for short-term predictions, so far they have not been able to outperform operational NWP
models (Weyn et al., 2020; Arcomano et al., 2020; Schultz et al., 2021). This might be, at least partly, due to the short training
sets provided by around 40 years of high-quality reanalysis data (Rasp and Thuerey, 2021). There are a number of ways to
tackle this problem, e.g., transfer learning could be used to blend data from low- and high-fidelity data/models (e.g., Ham et al.,
2019; Chattopadhyay et al., 2020e; Rasp and Thuerey, 2021), and/or physical constraints could be incorporated into the often
physics-agnostic ML models, which has been shown in applications of high-dimensional fluid dynamics (Raissi et al., 2020)
as well as toy examples of atmospheric or oceanic flows (Bihlo and Popovych, 2021). *The first contribution of this paper is to
provide a framework for the latter, based on building physical properties called equivariances into convolutional architectures
using deep spatial transformers that would specifically capture translation, rotation, and scaling.* The second contribution of
this paper is to equip these DDWP models with data assimilation (DA), which provides improved initial conditions for weather
forecasting and is one of the key reasons behind the success of NWP models. Below, we further discuss the need for integrating
DA and physical properties such as equivariances with DDWP models and briefly describe what has been already done in these
areas in previous studies.

Many of the DDWP models built so far are physics agnostic and learn the spatio-temporal evolution only from the training
data, resulting sometimes in physically inconsistent predictions and inability to capture key invariants and symmetries of the

| Term | Description |
|------|-------------|
| Autoregressive models | A model that iteratively predicts states of a system at new time steps by using the predicted state at the previous time step as input (Lütkepohl, 2013). |
| CNN | Convolutional neural network: a type of neural network in which features are learnt through successive convolutions and down-sampling of the input as it maps to the ouput (Goodfellow et al., 2016). |
| DDWP | Data-driven weather prediction: A framework in which a data-driven model is trained on historical weather data and predicts future weather without solving physical equations of the atmosphere (section 3.1). |
| DDWP+DA | Data-driven weather forecasting model as the background forecasting model integrated with SPEnKF DA algorithm (proposed in this paper in section 3.2). |
| Encoder-decoder | A neural network in which the input is encoded into a low-dimensioanl representation (encoding) and then decoded back into high-dimensional (often the same dimension as the input) space from the encoding (Goodfellow et al., 2016). |
| EnKF | Ensemble Kalman filter: A type of DA algorithm in which noisy observations from a system are ingested sequentially to incrementally provide better initial conditions for a dynamical model to predict the future states of the system (Evensen, 1994). |
| Equivariance | A property of a function that allows the output (of the function) to change appropriately in response to a transformation in the input (Wang et al., 2020; Bronstein et al., 2021). See section 3.1.2 for more details. |
| NWP | Numerical weather prediction |
| RNN | Recurrent neural network: A type of neural network in which information moves both forward and backwards through the network (Goodfellow et al., 2016). |
| SPEnKF | Sigma-point ensemble Kalman filter: A type of EnKF, in which ensembles are generated deterministically instead of randomly (Tang et al., 2014). |
| STN | Spatial transformer network: A neural network in which an affine transformation and subsequent interpolation allow the network to be equivariant (Jaderberg et al., 2015). |
| U-STN | U-NET with a spatial transformer connected to the latent space of the network (section 3.1.2) |
| Unscented transformation | A transformation that allows one to generate an optimal number of deterministic ensembles (Wan et al., 2001). In this paper, this transformation is used in SPEnKF; see section 3.2. |

**Table 1.** List of acronyms and technical ML and DA terms along with their brief descriptions.

underlying dynamical system, particularly when the training set is small (Reichstein et al., 2019; Chattopadhyay et al., 2020d). There are various approaches to incorporating some physical properties into the neural networks; for example, Kashinath et al. (2021) have recently reviewed 10 approaches (with examples) for physics-informed ML in the context of weather/climate modeling. One popular approach, in general, is to enforce key conservation laws, symmetries, or some (or even all) of the governing equations through custom-designed loss functions (e.g., Raissi et al., 2019; Beucler et al., 2019; Daw et al., 2020; Mohan et al., 2020; Thiagarajan et al., 2020; Beucler et al., 2021).

Another approach–which has received less attention particularly in weather/climate modeling–is to enforce the appropriate symmetries, which are connected to conserved quantities through the Noether's theorem (Hanc et al., 2004), inside the neural architecture. For instance, conventional CNN architectures enforce translational and rotational symmetries, which may not necessarily exist in the large-scale circulation; see Chattopadhyay et al. (2020d) for an example based on atmospheric blocking events and rotational symmetry. Indeed, recent research in the ML community has shown that preserving a more general property called "equivariance" can improve the performance of CNNs (Maron et al., 2018, 2019; Cohen et al., 2019). Equivariance-preserving neural network architectures learn the existence of (or lack thereof) symmetries in the data rather than enforcing them *a priori* and better track the relative spatial relationship of features (Cohen et al., 2019). In fact, in their work on forecasting midlatitude extreme-causing weather patterns, Chattopadhyay et al. (2020d) have shown that capsule neural networks, which are equivariance-preserving (Sabour et al., 2017), outperform conventional CNNs in terms of out-of-sample accuracy while requiring a smaller training set. Similarly, Wang et al. (2020) have shown the advantages of equivariance-preserving CNN architectures in data-driven modeling of Rayleigh-Bénard and ocean turbulence. More recently, using two-layer quasi-geostrophic turbulence as the test case, Chattopadhyay et al. (2020c) have shown that preserving equivariances related to translational, rotational, and scaling symmetry groups through a deep spatial transformer architecture (Jaderberg et al., 2015) improves the accuracy and stability of the DDWP models without increasing the network's complexity or computational cost (which are drawbacks of capsule neural networks). Building on these studies, here our first goal is to develop a physically consistent, autoregressive DDWP model that preserves equivariance using a deep spatial transformer in an encoder-decoder U-NET architecture (Ronneberger et al., 2015). Note that our approach to use a deep spatial transformer is different from enforcing invariants in the loss function in the form of partial differential equations of the system (Raissi et al., 2019).

DA is an essential component of modern weather forecasting (e.g., Kalnay, 2003; Carrassi et al., 2018; Lguensat et al., 2019). DA corrects the atmospheric state forecasted using a forward model (often a NWP model) by incorporating noisy and partial observations from the atmosphere (and other components of the Earth system), thus estimating a new corrected state of the atmosphere called "analysis", which serves as an improved initial condition for the forward model to forecast the future states. Most operational forecasting systems have their NWP model coupled to a DA algorithm that corrects the trajectory of the atmospheric states, e.g., every 6 h with observations from remote sensing and in-situ measurements. State-of-the-art DA algorithms use variational and/or ensemble-based approaches. The challenge with the former is computing the adjoint of the forward model, which involves high-dimensional, nonlinear partial differential equations (Penny et al., 2019). Ensemble-based approaches, which are usually variants of ensemble Kalman filter (EnKF, Evensen, 1994), bypass the need for computing the

adjoint but require generating a large ensemble of states that are each evolved in time using the forward model, which makes this approach computationally expensive (Hunt et al., 2007; Houtekamer and Zhang, 2016; Kalnay, 2003).

In recent years, there has been a growing number of studies at the intersection of ML and DA (Geer, 2021). A few studies have aimed to use ML to accelerate/improve DA frameworks, for example by taking advantage of their natural connection (Abarbanel et al., 2018; Kovachki and Stuart, 2019; Grooms, 2021; Hatfield et al., 2021). A few other studies have focused on using DA to provide suitable training data for ML from noisy/sparse observations (Brajard et al., 2020, 2021; Tang et al., 2020; Wikner et al., 2021). Others have integrated DA with a data-driven or hybrid forecast model for relatively simple dynamical systems (Hamilton et al., 2016; Lguensat et al., 2017; Lynch, 2019; Pawar and San, 2020). However, to the best of our knowledge, no study has yet integrated DA with a DDWP model. Here, our second goal is to present a DDWP+DA framework in which the DDWP is the forward model that efficiently provides a large, $O(1000)$, ensemble of forecasts for a sigma-point ensemble Kalman filter (SPEnKF) algorithm.

To provide proof-of-concepts for the DDWP model and the combined DDWP+DA framework, we use sub-daily 500 hPa geopotential height (Z500) from the ECMWF Reanalysis 5 (ERA5) dataset (Hersbach et al., 2020). The DDWP model is trained on hourly, 6 h, or 12 h Z500 samples. The spatio-temporal evolution of Z500 is then forecasted from precise initial conditions using the DDWP model or from noisy initial conditions using the DDWP+SPEnKF framework. Our main contributions in this paper are three-fold, namely:

- Introducing the equivariance-preserving encoder-decoder U-NET with a deep spatial transformer architecture for DDWP modeling and showing the advantages of this architecture over a conventional encoder-decoder U-NET.

- Introducing the DDWP+DA framework, which leads to stable DA cycles without the need for any localization or inflation by taking advantage of the large forecast ensembles produced in a data-driven fashion using the DDWP model.

- Introducing a novel multi-time-step method for improving the DDWP+DA framework. This framework utilizes virtual observations produced using more accurate DDWP models that have longer time steps. This framework exploits the non-trivial dependence of the accuracy of autoregressive data-driven models on the time step size.

The remainder of the paper is structured as follows. The data are described in Section 2. The encoder-decoder U-NET architecture with the deep spatial transformer and the SPEnKF algorithm are introduced in Section 3. Results are presented in Section 4 and the Discussion and Summary are in Section 5.

## 2 Data

We use the ERA5 dataset from the WeatherBench repository (Rasp et al., 2020), where each global sample of Z500 at every hour is downsampled to a rectangular longitude-latitude $(x, y)$ grid of $32 \times 64$. We have chosen the variable Z500 following previous work (Weyn et al., 2019, 2020; Rasp et al., 2020) as an example, because it is representative of the large-scale circulation in the troposphere and influences near-surface weather and extremes. This coarse-resolution Z500 dataset from the WeatherBench repository has been used in a number of recent studies to perform data-driven weather forecasting (Rasp et al.,

2020; Rasp and Thuerey, 2021). Here, we use Z500 data from 1979 to 2015 ($\approx 315360$ samples) for training, 2016–2017 ($\approx 17520$ samples) for validation, and 2018 ($\approx 8760$ samples) for testing.

## 120  3  Methods

### 3.1  The equivariance-preserving DDWP model: U-NET with a deep spatial transformer (U-STN)

The DDWP models used in this paper are trained on Z500 data without access to any other atmospheric fields that might affect the atmosphere's spatio-temporal evolution. Once trained on past Z500 snapshots sampled at every $\Delta t$, the DDWP model takes Z500 at a particular time $t$ ($Z(t)$ hereafter) as the input and predicts $Z(t+\Delta t)$, which is then used as the input to predict 125  $Z(t+2\Delta t)$, and this autoregressive process continues as needed. We use $\Delta t$ that is 1, 6, or 12 h. The baseline DDWP model used here is a U-NET similar to the one used in Weyn et al. (2020). For the DDWP introduced here, the encoded latent space of the U-NET is coupled with a deep spatial transformer (U-STN hereafter) to preserve equivariance between the latent space of the network and the decoded output. The preservation of equivariance enables the U-STN to track translation, rotation, and stretching of the synoptic- and larger-scale patterns, and is expected to improve the forecast of the spatio-temporal evolution of 130  the midlatitude Rossby waves and their nonlinear breakings. In this section, we briefly discuss the U-STN architecture, which is schematically shown in Fig. 1. Note that from now on, "x" in U-STNx (and U-NETx) indicates the $\Delta t$ that is used, e.g., U-STN6 uses $\Delta t = 6$ h.

#### 3.1.1  Localization network or encoding block of U-STN

The network takes in an input snapshot of Z500, $Z(t)^{32\times64}$, as initial condition and projects it onto a low-dimensional encoding 135  space via a U-NET convolutional encoding block. This encoding block performs two successive sets of two convolution operations (without changing the spatial dimensions) followed by a max-pooling operation. It is then followed by two convolutions without max-pooling and four dense layers. More details on the exact set of operations inside the architecture are reported in Table 2. The convolutions inside the encoder block account for Earth's longitudinal periodicity by performing circular convolutions (Schubert et al., 2019) on each feature map inside the encoder block. The encoded feature map, which is the output of 140  the encoding block and consists of the reduced $Z$ and co-ordinate system, $\tilde{Z}^{8\times16}$ and $(x^o_i, y^o_i)$ where $i = 1, 2 \ldots 8 \times 16$, is sent to the spatial transformer module described below.

#### 3.1.2  Spatial transformer module

The spatial transformer (Jaderberg et al., 2015) applies an affine transformation $T(\theta)$ to the reduced co-ordinate system $(x^o_i, y^o_i)$ to obtain a new transformed co-ordinate system $(x^s_i, y^s_i)$:

145
$$\begin{bmatrix} x^s_i \\ y^s_i \end{bmatrix} = T(\theta) \begin{bmatrix} x^o_i \\ y^o_i \\ 1 \end{bmatrix}, \tag{1}$$

where

$$T(\theta) = \begin{bmatrix} \theta_{11} & \theta_{12} & \theta_{13} \\ \theta_{21} & \theta_{22} & \theta_{23} \end{bmatrix}.$$ (2)

The parameters $\theta$ are predicted for each sample. A differentiable sampling kernel (a bi-linear interpolation kernel in this case) is then used to transform $\tilde{Z}^{8 \times 16}$, which is on the old co-ordinate system $(x_i^o, y_i^o)$, into $\bar{Z}^{8 \times 16}$, which is on the new co-ordinate system $(x_i^s, y_i^s)$. The spatial transformer module ensures that the transformation in the encoded latent space is equivariance-preserving (Esteves et al., 2018). Note that in this architecture, the spatial transformer is applied to the latent space and its objective is to ensure that no *a priori* symmetry structure is assumed in the latent space. The parameters in $T(\theta)$ learn the transformation (translation, rotation, and scaling) between the input to the latent space and the decoded output. It must be noted here, that this does not ensure that the entire network is equivariant by construction.

We highlight that in this paper, we are focusing on capturing effects of translation, rotation, and scaling of the input field, because those are the ones that we expect to matter the most for the synoptic patterns on a 2D plane. Other transformations and equivariance groups could be similarly included (Wang et al., 2020). Furthermore, here we focus on an architecture with a transformer that acts only on the latent space. More complex architectures, with transformations like Eq. (1) after every convolution layer, can be used too albeit with a significant increase in computational cost (de Haan et al., 2020; Wang et al., 2020). Our preliminary exploration shows that for this work, the one spatial transformer module applied on the latent space of the U-NET yields sufficiently superior performance (over the baseline, U-NET), but further exhaustive explorations should be conducted in future studies to find the best performing architecture for each application. Moreover, recent work in neural architecture search for geophysical turbulence shows that, with enough computing power, one can perform exhaustive searches over optimal architectures, a direction that should be pursued in future work (Maulik et al., 2020).

Finally we point out that without the transformer module, $\bar{Z} = \tilde{Z}$, and the network becomes a standard U-NET.

### 3.1.3 Decoding block

The decoding block is a series of deconvolution layers (convolution with zero-padded upsampling) concatenated with the corresponding convolution outputs from the encoder part of the U-NET. The decoding blocks bring the latent space $\bar{Z}^{8 \times 16}$ back into the original dimension and co-ordinate system at time $t + \Delta t$, thus outputting $Z(t + \Delta t)^{32 \times 64}$. The concatenation of the encoder and decoder convolution outputs allows the architecture to learn the features in the small-scale dynamics of Z500 better (Weyn et al., 2020).

The loss function $L$ to be minimized is

$$L(\lambda) = \frac{1}{(N+1)} \sum_{t=0}^{t=N\Delta t} ||\,(Z(t + \Delta t) - \text{U-STNx}\,(Z(t), \lambda))\,||_2^2,$$ (3)

where $N$ is the number of training samples, $t = 0$ is the start time of the training set, and $\lambda$ represents the parameters of the network that are to be trained (in this case, the weights, biases, and $\theta$ of U-STNx). In both encoding and decoding blocks, the ReLU activation functions are used. The number of convolutional kernels (32 in each layer), size of each kernel ($5 \times 5$),

Gaussian initialization, and the learning rate ($\alpha = 3 \times 10^{-4}$) have been chosen after extensive trial-and-error. All codes for these networks (as well as DA) have been made publicly available on GitHub and Zenodo (see the Code Availability statement). A comprehensive list of information about each of the layers in both the U-STNx and U-NETx architectures is presented in Table. 2 along with the optimal set of hyperparameters that have been obtained through extensive trial-and-error.

Note that, the use of U-NET is inspired from the works of Weyn et al. (2020), however, the architecture used in this study is different from that of Weyn et al. (2020). The main differences are in the number of convolution layers and filters used in the U-NET along with the spatial transformer module. Apart from that, in Weyn et al. (2020), the mechanism by which autoregressive prediction is done, is different from this paper. Two time steps (6h and 12h) are predicted directly as the output by Weyn et al. (2020) using the U-NET. Moreover, the data for training and testing in Weyn et al. (2020) are on the gnomonic cubed sphere.

## 3.2 Data assimilation algorithm and coupling with DDWP

For DA, we employ the SPEnKF algorithm, which unlike the EnKF algorithm, does not use random perturbations to generate an ensemble but rather uses an unscented transformation (Wan et al., 2001) to deterministically find an optimal set of points called sigma points (Ambadan and Tang, 2009). The SPEnKF algorithm has been shown to outperform EnKF on particular test cases for both chaotic dynamical systems and ocean dynamics (Tang et al., 2014) although whether it is always superior to EnKF is a matter of active research (Hamill et al., 2009) and beyond the scope of this paper. Our DDWP+DA framework can use any ensemble-based algorithm.

In the DDWP+DA framework, shown schematically in Fig. 2, the forward model is a DDWP, which is chosen to be U-STN1 and denoted as $\mathbf{\Psi}$ below. We use $\sigma_{\text{obs}}$ for the standard deviation of the observation noise, which in this paper is either $\sigma_{\text{obs}} = 0.5\sigma_Z$ or $\sigma_{\text{obs}} = \sigma_Z$, where $\sigma_Z$ is the standard deviation of $Z500$ over all grid points and over all years between 1979–2015. Here, we assume that the noisy observations are assimilated every 24 h (again, the framework can be used with any DA frequency, such as 6 h, which is used commonly in operational forecasting).

We start with a noisy initial condition $Z(t)$, and use U-STN1 to autoregressively (with $\Delta t = 1$ h) predict the next time steps, $Z(t + \Delta t)$, $Z(t + 2\Delta t)$, $Z(t + 3\Delta t)$, up to $Z(t + 23\Delta t)$. For a $D$-dimensional system (i.e., $Z \in \mathbb{R}^D$), the optimal number of ensemble members for SPEnKF is $2D$ (Ambadan and Tang, 2009). Because here $D = 32 \times 64$, 4096 ensemble members are needed. While this is a very large ensemble size if the forward models is a NWP (operationally, $\sim 50 - 100$ members are used (Leutbecher, 2019)), the DDWP can inexpensively generate $O(1000)$ ensemble members, a major advantage of DDWP as a forward model that we will discuss later in Section 5.

To do SPEnKF, an ensemble of states at the $23^{\text{rd}}$ hour of each DA cycle (24 h is one DA cycle) is generated using a symmetric set of sigma points (Julier and Uhlmann, 2004) as

$$
\begin{aligned}
Z^i_{\text{ens}}(t + 23\Delta t) &= Z(t + 23\Delta t) - A_i, \\
Z^j_{\text{ens}}(t + 23\Delta t) &= Z(t + 23\Delta t) + A_j,
\end{aligned}
\tag{4}
$$

| Layer number | Layer type | Kenel size | number of filters/number of neurons | Output Size | Activation |
|:---:|:---:|:---:|:---:|:---:|:---:|
| 1 | Convolution | $5 \times 5$ | 32 | $32 \times 64$ | ReLU |
| 2 | Convolution | $5 \times 5$ | 32 | $32 \times 64$ | ReLU |
| 3 | Max Pooling ($2 \times 2$) | – | – | $16 \times 32$ | – |
| 4 | Convolution | $5 \times 5$ | 32 | $16 \times 32$ | ReLU |
| 5 | Convolution | $5 \times 5$ | 32 | $16 \times 32$ | ReLU |
| 6 | Max Pooling ($2 \times 2$) | – | – | $8 \times 16$ | – |
| 7 | Convolution | $5 \times 5$ | 32 | $8 \times 16$ | ReLU |
| 8 | Convolution | $5 \times 5$ | 32 | $8 \times 16$ | ReLU |
| 9 | Fully connected | – | 500 | 500 | ReLU |
| 10 | Fully connected (only for STN) | – | 200 | 200 | ReLU |
| 11 | Fully connected (only for STN) | – | 100 | 100 | ReLU |
| 12 | Fully connected (only for STN) | – | 50 | 50 | ReLU |
| 13 | Affine transformation (only for STN) | – | – | – | – |
| 14 | Bi-linear interpolation (only for STN) | – | – | $8 \times 16$ | – |
| 15 | Up-sampling + concatenate Layer 5 | – | – | $16 \times 32$ | – |
| 16 | Convolution | $5 \times 5$ | 32 | $16 \times 32$ | ReLU |
| 17 | Convolution | $5 \times 5$ | 32 | $16 \times 32$ | ReLU |
| 18 | Up-sampling + concatenate layer 2 | – | – | $32 \times 64$ | – |
| 19 | Convolution | $5 \times 5$ | 32 | $32 \times 64$ | ReLU |
| 20 | Convolution | $5 \times 5$ | 32 | $32 \times 64$ | ReLU |
| 21 | Convolution | $5 \times 5$ | 32 | $32 \times 64$ | Linear |

**Table 2.** Table presenting information on the U-STNx and U-NETx architecture with the optimal set of hyperparameters that have been obtained after extensive trial and error. Note that apart from the affine transformation and bi-linear interpolation layer, the U-STNx and U-NETx architecture are similar. The networks have been implemented in Tensorflow and Keras.

where $i, j \in [1, 2, \cdots D = 32 \times 64]$ are indices of the $2D$ ensemble members. Vectors $A_i$ and $A_j$ are columns of matrix $\mathbf{A} = \mathbf{U}\sqrt{\mathbf{S}}\mathbf{U^T}$, where $\mathbf{U}$ and $\mathbf{S}$ are obtained from the singular value decomposition of the analysis covariance matrix $\mathbf{P_a}$, i.e., $\mathbf{P_a} = \mathbf{USV}^T$. The $D \times D$ matrix $\mathbf{P_a}$ is either available from the previous DA cycle (see Eq. (10) below) or is initialized as an identity matrix at the beginning of DA. Note that here, we generate the ensemble at one $\Delta t$ before the next DA; however, the ensembles can be generated at any time within the DA cycle and carried forward although that would increase the computational cost of the framework. We have explored generating the ensembles at $t + 0\Delta t$ (i.e., the beginning) but did not find any improvement over Eq. (4). It must however be noted that by not propagating the ensembles for $24$ h the spread of the ensembles underestimates the background error.

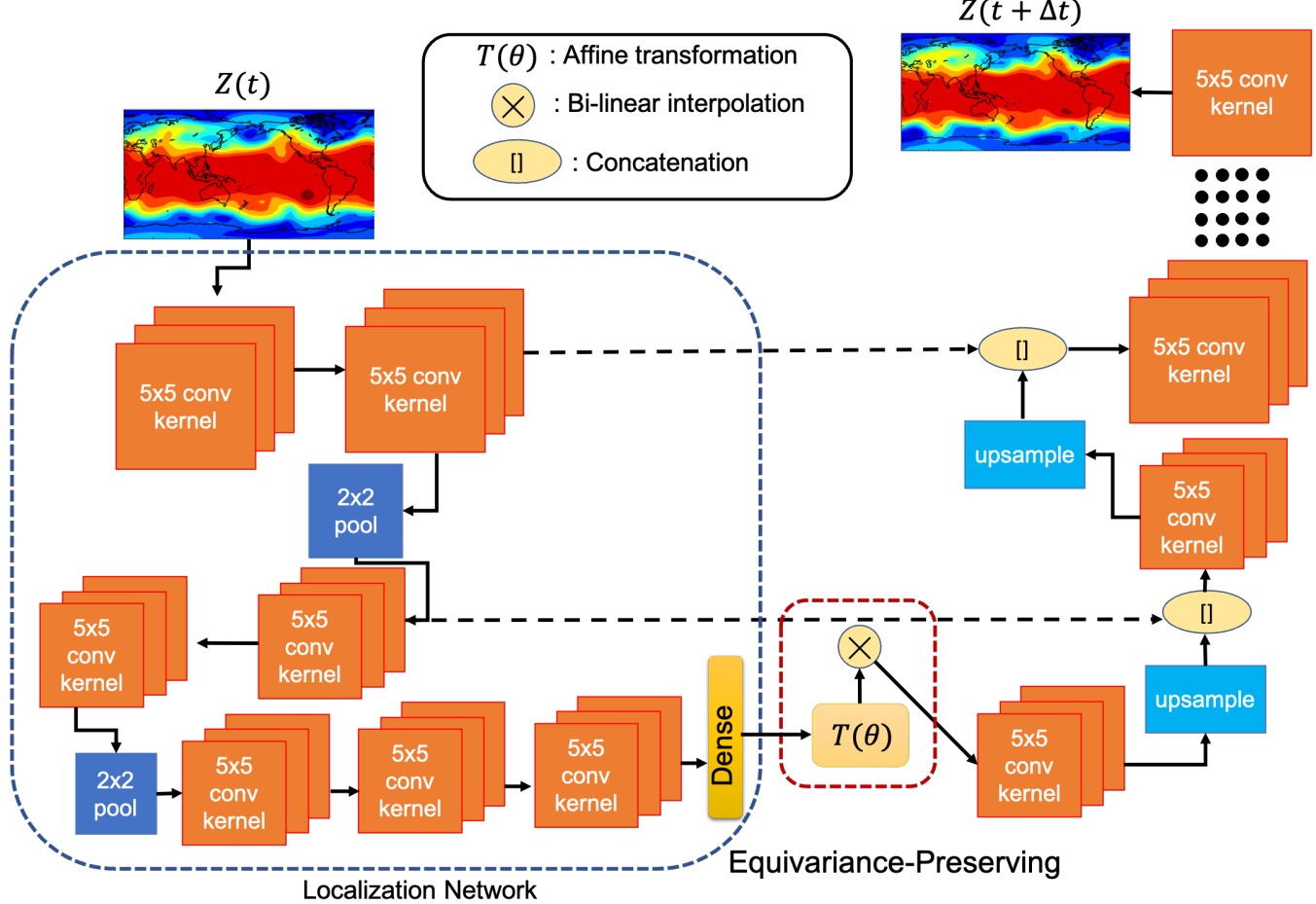

**Figure 1.** A schematic of the architecture of U-STNx. The architecture preserves equivariance between the input to the latent space and the decoded output owing to the spatial transformer module implemented through the affine transformation, $T(\theta)$, along with the differentiable bi-linear interpolation kernel. The network integrates $Z(t)$ to $Z(t+\Delta t)$. The size of the bottleneck layer is $8 \times 16$. *Note that the schematic does not show the exact number of layers and number of filters used in U-STNx and U-NETx for the sake of clarity. The information on the number of layers and number of filters along with the activation function used is shown in Table 2.*

Once the ensembles are generated via Eq. (4), every ensemble member is fed into $\mathbf{\Psi}$ to predict an ensemble of forecasted states at $t + 24\Delta t$:

$$Z^k_{\text{ens}}(t + 24\Delta t) = \mathbf{\Psi}\left(Z^k_{\text{ens}}(t + 23\Delta t)\right), \tag{5}$$

where $k \in \{-D, -D+1, \cdots, D-1, D\}$. In general, the modeled observation is $\mathbf{H}\left(\langle Z^k_{\text{ens}}(t + 24\Delta t)\rangle, \epsilon(t)\right)$, where $\mathbf{H}$ is the observation operator and $\epsilon(t)$ is the Gaussian random process with standard deviation $\sigma_{\text{obs}}$ that represents the uncertainty in the observation. $\langle . \rangle$ denotes ensemble averaging. In this paper, we assume that $\mathbf{H}$ is the identity matrix while we acknowledge

220

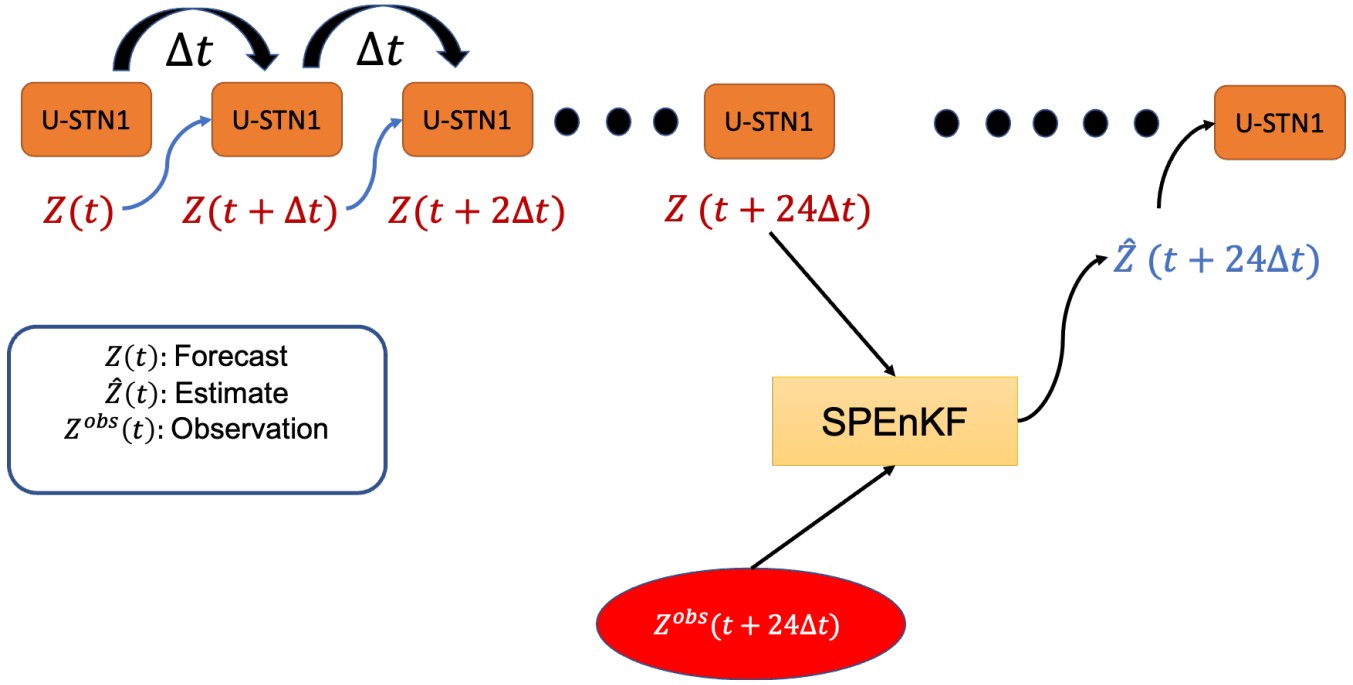

**Figure 2.** The framework for a synergistic integration of a DA algorithm (SPEnKF) with a DDWP (U-STN1). Once the DDWP+DA framework is provided with a noisy $Z(t)$, it uses U-STN1 to autoregressively predict $Z(t + 23\Delta t)$. A large ensemble is then generated using Eq. (4), and for each member $k$, $Z_{\text{ens}}^k(t + 24\Delta t)$ is predicted using U-STN1. Following that, an SPEnKF algorithm assimilates a noisy observation at the $24^{\text{th}}$ h to provide the estimate (analysis) state of Z500, $\hat{Z}(t + 24\Delta t)$. U-STN1 then uses this analysis state as the new initial condition and evolves the state in time, with DA occurring every 24 hours.

that in general, it could be a nonlinear function. The SPEnKF algorithm can account for such complexity, but here, to provide a proof-of-concept, we have assumed that we can observe the state, although with a certain level of uncertainty. With $\mathbf{H} = \mathbf{I}$, the background error covariance matrix $\mathbf{P_b}$ becomes

$$\mathbf{P_b} = \mathbf{E}\left[\left(Z_{\text{ens}}^k(t + 24\Delta t) - \langle Z_{\text{ens}}^k(t + 24\Delta t)\rangle\right)\left(Z_{\text{ens}}^k(t + 24\Delta t) - \langle Z_{\text{ens}}^k(t + 24\Delta t)\rangle\right)^T\right], \tag{6}$$

where $[.]^T$ denotes the transpose operator and $\mathbf{E}[.]$ denotes the expectation operator. The innovation covariance matrix is defined as:

$$\mathbf{C} = \mathbf{P_b} + \mathbf{R}, \tag{7}$$

where the observation noise matrix $\mathbf{R}$ is a constant diagonal matrix of the variance of observation noise, i.e., $\sigma_{\text{obs}}^2$. The Kalman gain matrix is then given by

$$\mathbf{K} = \mathbf{P_b}\mathbf{C}^{-1}, \tag{8}$$

and the estimated (analysis) state $\hat{Z}(t+24\Delta t)$ is calculated as

$$\hat{Z}(t+24\Delta t) = \langle Z(t+24\Delta t)\rangle - \mathbf{K}\left(\langle Z_{\text{ens}}^k(t+24\Delta t)\rangle - Z^{\text{obs}}(t+24\Delta t)\right), \tag{9}$$

where $Z^{\text{obs}}(t+24\Delta t)$ is the noisy observed Z500 at $t+24\Delta t$; i.e., ERA5 value at each grid point plus random noise drawn from
$\mathcal{N}(0, \sigma_{\text{obs}}^2)$. While adding Gaussian random noise to the truth is an approximation, it is a quite common in the DA literature
(Brajard et al., 2020, 2021; Pawar et al., 2020). The analysis error covariance matrix is updated as

$$\mathbf{P_a} = \mathbf{P_b} - \mathbf{KCK^T}. \tag{10}$$

The estimated state $\hat{Z}(t+24\Delta t)$ becomes the new initial condition to be used by U-STN1 and the updated $\mathbf{P_a}$ is used to
generate the ensembles in Eq. (4) after another 23 h for the next DA cycle.

Finally, we remark that often with low ensemble sizes, the background covariance matrix, $\mathbf{P_b}$ (Eq. (6)), suffers from spurious
correlations which are corrected using localization and inflation strategies (Hunt et al., 2007; Asch et al., 2016). However, due to
the large ensemble size used here (with 4096 ensemble members that are affordable because of the computationally inexpensive
DDWP forward model) we do not need to perform any localization or inflation on $\mathbf{P_b}$ to get stable DA cycles as shown in the
next section.

## 4  Results

### 4.1  Performance of physically consistent DDWP: Noise-free initial conditions (no DA)

First, we compare the performance between a U-STN and a conventional U-NET, whose only difference is in the use of the
spatial transformer module in the former. Using U-STN12 and U-NET12 as representatives of these architectures, Fig. 3 shows
the anomaly correlation coefficients (ACCs) between the predictions from U-STN12 or U-NET12 and the truth (ERA5) for 30
noise-free, random initial conditions. ACC is computed every 12 h as the correlation coefficient between the predicted Z500
anomaly and the Z500 anomaly of ERA5, where anomalies are derived by removing the 1979-2015 time mean of Z500 of the
ERA5 dataset. U-STN12 clearly outperforms U-NET12, most notably after 36 h, reaching ACC=0.6 after around 132 h, a $45\%$
(1.75 day) improvement over U-NET12, which reaches ACC=0.6 after around 90 h.

To further see the source of this improvement, Fig. 4 shows the spatio-temporal evolution of Z500 patterns from an example
of prediction using U-STN12 and U-NET12. Comparing with the truth (ERA5), U-STN12 can better capture the evolution
of the large-amplitude Rossby waves and the wavebreaking events compared to U-NET12; e.g., see the patterns over Central
Asia, Southern Pacific Ocean, and Northern Atlantic Ocean on days 2-5. We cannot rigorously attribute the better capturing
of wavebreaking events to an improved representation of physical features by the spatial transformer. However, the overall
improvement in performance of U-STN12 due to the spatial transformer (which is the only difference between U-STN12 and
U-NET12) may lead to capturing some wavebreaking events in the atmosphere as can be seen from exemplary evidence in
Fig. 4. Furthermore, on days 4 and 5, the predictions from U-NET12 have substantially low Z500 values in the high latitudes
of the Southern Hemisphere, showing signs of unphysical drifts.

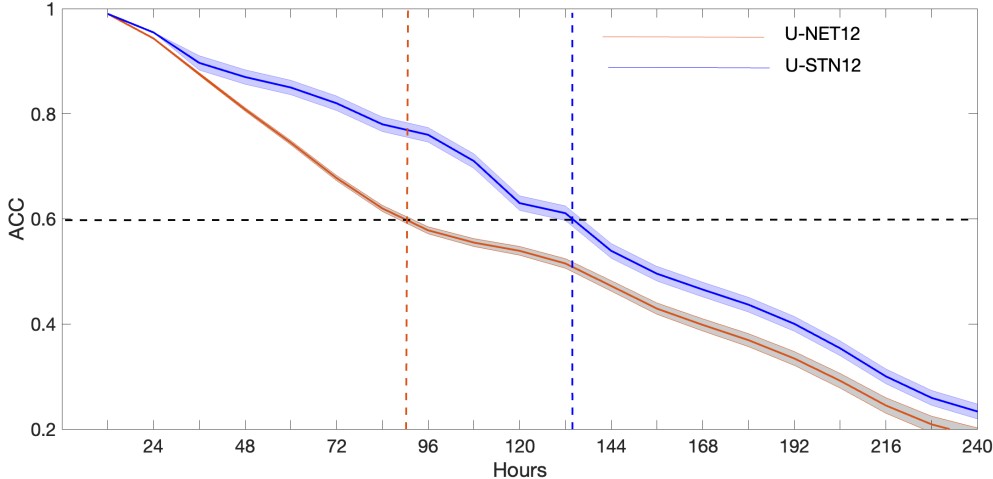

**Figure 3.** Anomaly correlation coefficient (ACC) calculated between Z500 anomalies of ERA5 and Z500 anomalies predicted using U-STN12 or U-NET12 from 30 noise-free, random initial conditions. The solid lines and the shadings show the mean and the standard deviation over the 30 initial conditions.

Overall, the results of Figs. 3 and 4 show the advantages of using the spatial transformer enabled U-STNs in DDWP models. It is important to note that it is difficult to assert whether the transformation with $T(\theta)$ in the latent space actually leads to physically meaningful transformations in the decoded output. However, we see that the performance of the network improves with the addition of the spatial transformer module. Future studies need to focus on more interpretation of what the $T(\theta)$ matrix inside neural networks capture (Bronstein et al., 2021). Note that while here we show results with $\Delta t = 12$ h, similar improvements are seen with $\Delta t = 1$ h and $\Delta t = 6$ h (see section 4.3). Furthermore, to provide a proof-of-concept for the U-STN, in this paper we focus on Z500 (representing the large-scale circulation) as the only state variable to be learnt and predicted. Even without access to any other information (for example about small scales), the DDWP model can provide skillful forecasts for some time, consistent with earlier findings with the multi-scale Lorenz 96 system (Dueben and Bauer, 2018; Chattopadhyay et al., 2020b). More state variables can be easily added to the framework, which is expected to extend the forecast skills, based on previous work with U-NETs (Weyn et al., 2020). In this work, we have considered Z500 as an example for a proof-of-concept. We have also performed experiments (not shown for brevity) by adding T850 as one of the variables to the input along with Z500 in U-NETx and U-STNx and found similarly good prediction performance for the T850 variable.

A benchmark for different DDWP models has been shown in Rasp et al. (2020), with different ML algorithms such as CNN, linear regression, etc. In terms of RMSE for Z500 (Fig. 6, left panel, shows RMSE of U-STNx and U-NETx in this paper with different $\Delta t$), U-STN12 outperforms the CNN model in WeatherBench (Rasp et al., 2020) by 33.2 m at lead time of 3 days, and 26.7 m at lead time of 5 days. Similarly, U-STN12 outperforms the linear regression in WeatherBench by 39.9 m at lead

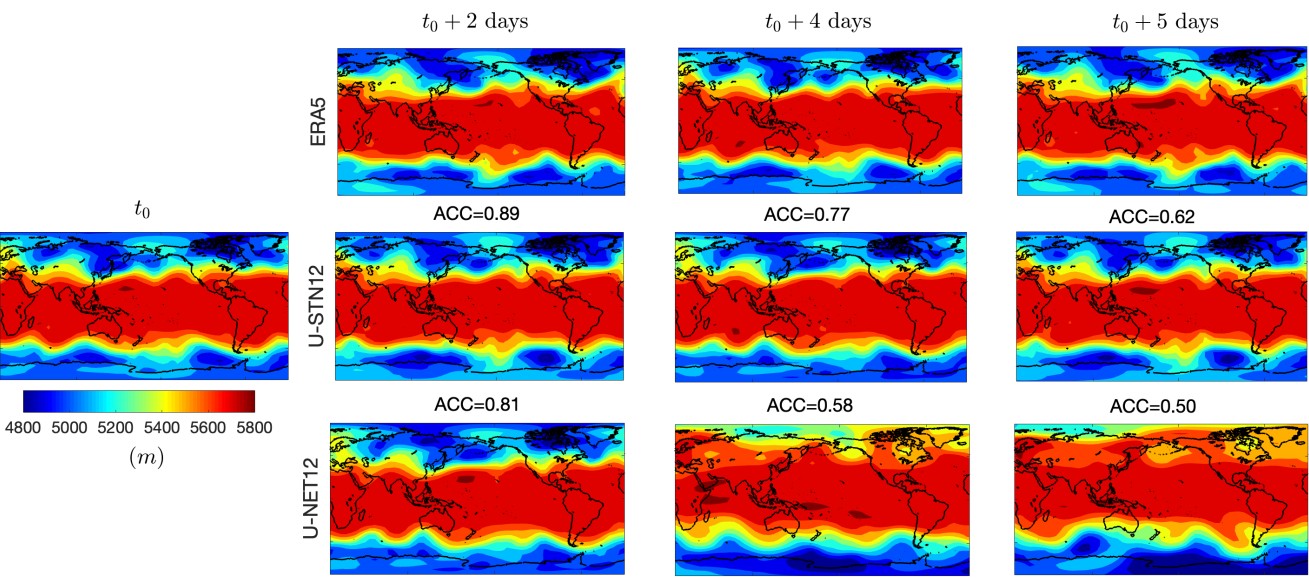

**Figure 4.** Examples of the spatio-temporal evolution of Z500 predicted from a noise-free initial condition ($t_0$) using U-STN12 and U-NET12, and compared with the truth from ERA5. For the predicted patterns, the anomaly correlation coefficient (ACC) is shown above each panel (see the text for details).

time of 3 days and by 29.3 m at lead time of 5 days. Note that in more recent work (Weyn et al., 2020; Rasp and Thuerey, 2021), prediction horizons outperforming the WeatherBench models (Rasp et al. (2020)) have also been shown.

### 4.2 Performance of the DDWP+DA framework: noisy initial conditions and assimilated observations

To analyse the performance of the DDWP+DA framework, we use U-STN1 as the DDWP model and SPEnKF as the DA algorithm, as described in Section 3.2. In this U-STN1+SPEnKF setup, the initial conditions for predictions are noisy observations and every 24 h, noisy observations are assimilated to correct the forecast trajectory (as mentioned before, noisy observations are generated by adding random noise from $\mathcal{N}(0, \sigma_{obs})$ to the Z500 of ERA5).

In Fig. 5, for 30 random initial conditions and two noise levels ($\sigma_{obs} = 0.5\sigma_Z$ or $1\sigma_Z$), we report the spatially averaged root-mean-squared-error (RMSE) and the correlation coefficient (R) of the forecasted full Z500 fields as compared to the truth, i.e., the (noise-free) Z500 fields of ERA5. For both noise levels, we see that within each DA cycle, the forecast accuracy decreases between 0 and 23 h until DA with SPEnKF occurs at the $24^{th}$ hour wherein information from the noisy observation is assimilated to improve the estimate of the forecast at the $24^{th}$ hour. This estimate acts as the new improved initial condition to be used by U-STN1 to data drivenly forecast future time steps. In either case, the RMSE and R remain below 30 m (80 m) and above 0.7 (0.3) with $\sigma_{obs} = 0.5\sigma_Z$ ($\sigma_{obs} = 1\sigma_Z$) for the first 10 days. The main point here is not the accuracy of the forecast

(which as mentioned before, could be further extended, for example by adding more state variables), but the stability of the U-STN1+SPEnKF framework (without localization/inflation), which even with the high noise level, can correct the trajectory, and increase R from $\sim 0.3$ to $0.8$ in each cycle. Although not shown in this paper, the U-STN1+SPEnKF framework remains stable beyond the 10 days and shows equally good performance for longer periods of time.

One last point to make here is that within each DA cycle, the maximum forecast accuracy is not at when DA occurs, but

3-4 h later (this is most clearly seen for the case with $\sigma_{obs} = 1\sigma_Z$ in Fig. 5). A likely reason behind the further improvement of the performance after DA is the de-noising capability of neural networks when trained on non-noisy training data (Xie et al., 2012).

### 4.3 DDWP+DA with virtual observations: A multi-time-step framework

One might wonder how the performance of the DDWP model (with or without DA) depends on $\Delta t$. Figure 6 compares the

305 performance of U-STNx as well as U-NETx for $\Delta t = 1$, 6, and 12 h for 30 random noise-free initial conditions (no DA). It is clear that the DDWP models with larger $\Delta t$ outperform the ones with smaller $\Delta t$; i.e., in terms of forecast accuracy, U-STN12 > U-STN6 > U-STN1. This trends holds true for both U-STNx and U-NETx, while as discussed before, for the same $\Delta t$, the U-STN outperforms the U-NET.

This dependence on $\Delta t$ might seem counter-intuitive as it is opposite of what one sees in numerical models, whose forecast

errors decrease with smaller time steps. The increase in the forecast errors of these DDWP models when $\Delta t$ is decreased is likely due to the non-additive nature of the error accumulation of these autoregressive models. The data-driven models have some degree of generalization error (for out-of-sample prediction), and every time the model is invoked to predict the next time step, this error is accumulated. For neural networks, this accumulation is not additive and propagates nonlinearly during the autoregressive prediction. Currently, these error propagations are not understood well enough to build a rigorous framework

for estimating the optimal $\Delta t$ for data-driven, autoregressive forecasting; however, this behavior has been reported in other studies on nonlinear dynamical systems and can be exploited to formulate multi-time-step data-driven models; see (Liu et al., 2020) for an example (though without DA).

Based on the trends seen in Fig. 6, we propose a novel idea for a multi-time-step DDWP+DA framework, in which the forecasts from the more accurate DDWP with larger $\Delta t$ are incorporated as virtual observations, using DA, into the forecasts

of the less accurate DDWP with smaller $\Delta t$, thus providing overall more accurate short-term forecasts. Figure 7 shows a schematic of this framework for the case where the U-STN12 model provides the virtual observations that are assimilated using the SPEnKF algorithm in the middle of the 24 h DA cycles into the hourly forecasts from U-STN1. At $24^{th}$ hour, noisy observations are assimilated using the SPEnKF algorithm as before.

Figure 8 compares the performance of the multi-time-step U-STNx+SPEnKF framework, which uses virtual observations

from U-STN12, with that of U-STN1+SPEnKF, which was introduced in Section 4.2, for the case with $\sigma_{obs} = 0.5\sigma_Z$. In terms of both RMSE and R, the multi-time-step U-STNx+SPEnKF framework outperforms the U-STN1+SPEnKF framework, as for example, the maximum RMSE of the former is often comparable to the minimum RMSE of the latter. Figure 9 shows the same

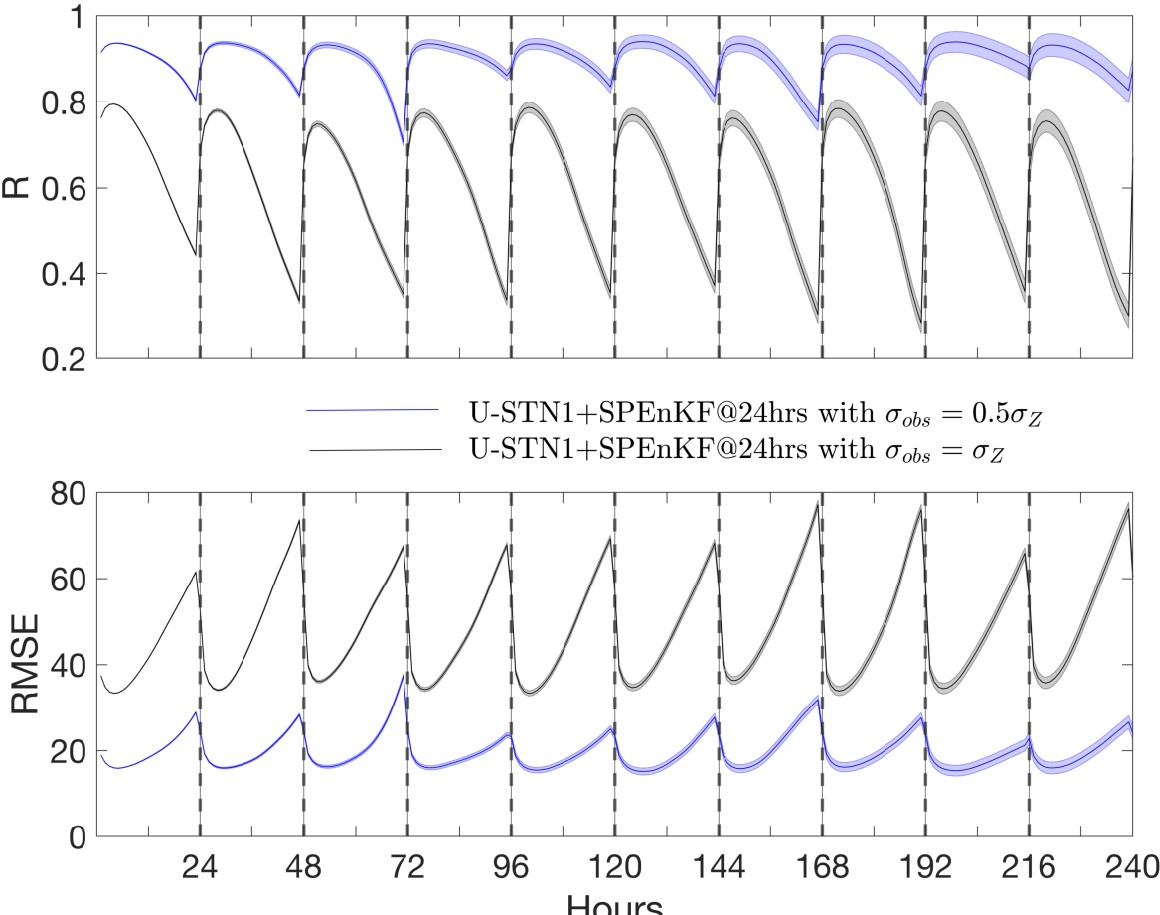

**Figure 5.** The top panel shows R and the bottom panel shows RMSE (in meters) between noise-free data from ERA5 and the forecasts from U-STN1+SPEnKF for two levels of observation noise. Predictions are started from 30 random noisy observations. The lines (shading) show the mean (standard deviation) of the 30 forecasts. Noisy observations are assimilated every 24 h (indicated by black, dashed vertical lines).

analysis but for the case with larger observation noise $\sigma_{\text{obs}} = \sigma_Z$, which further demonstrates the benefits of the multi-time-step framework and use of virtual observations.

The multi-time-step framework with assimilated virtual observations introduced here improves the forecasts of short-term intervals by exploiting the non-trivial dependence of the accuracy of autoregressive, data-driven models on time step size. While hourly forecasts of Z500 may not be necessarily of practical interest, the framework can be applied in general to any state variable, and can be particularly useful for multi-scale systems with a broad range of spatio-temporal scales. A similar

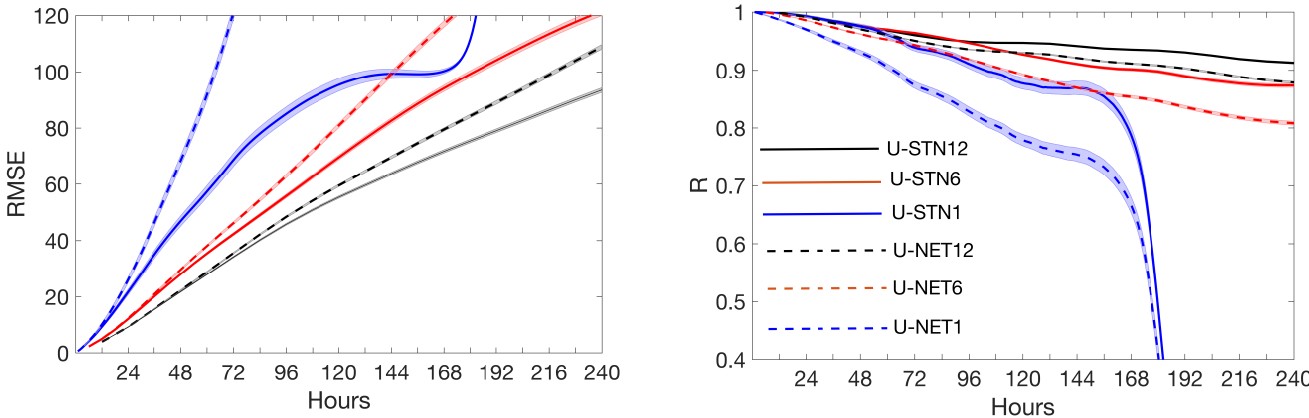

**Figure 6.** The left (right) panel shows RMSE (R) between noise-free data from ERA5 and the forecasts from U-STNx or U-NETx from 30 random, noise-free initial conditions. No DA is used here. RMSE is in meters. The lines (shading) show the mean (standard deviation) of the 30 forecasts.

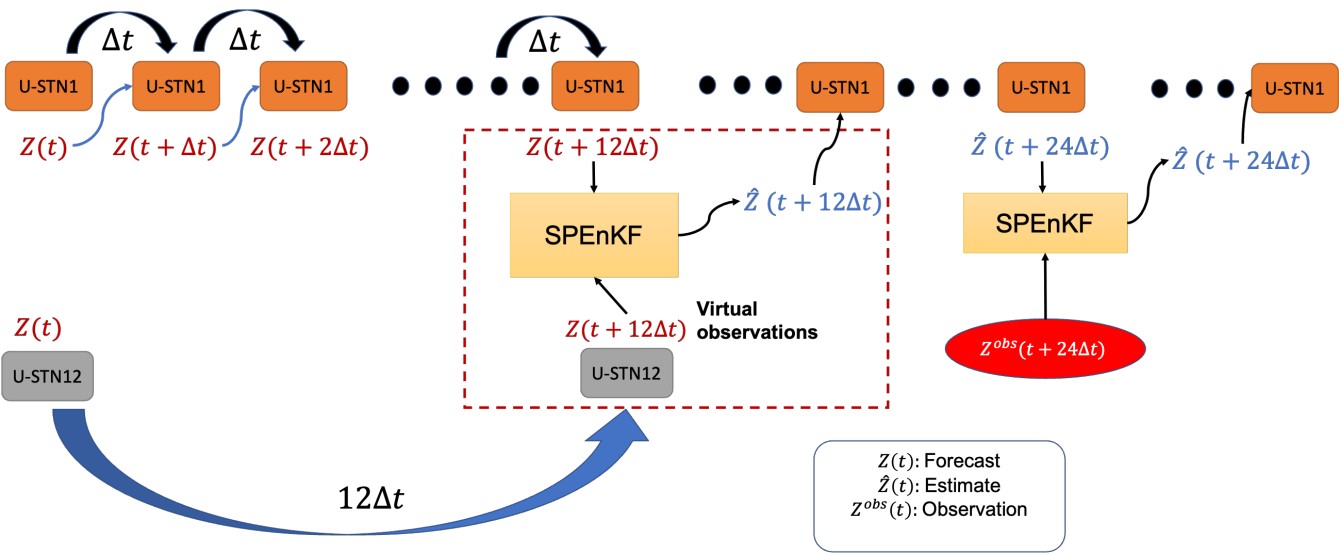

**Figure 7.** Schematic of the multi-time-step DDWP+DA framework. The U-STN12 model provides forecasts every 12 h, which are assimilated as virtual observations using SPEnKF into the U-STN1+SPEnKF framework that has a 24 h DA cycle for assimilating noisy observations. At 12[th] hours, the U-STN12 forecasts are more accurate than those from the U-STN1 model, enabling the framework to improve the prediction accuracy every 12[th] hour, thereby improving the initial condition used for the next forecasts before DA with noisy observations (every 24 h).

idea was used in Bach et al. (2021), wherein data-driven forecasts of oscillatory modes with singular spectrum analysis and an analog method were used as virtual observations to improve the prediction of a chaotic dynamical system.

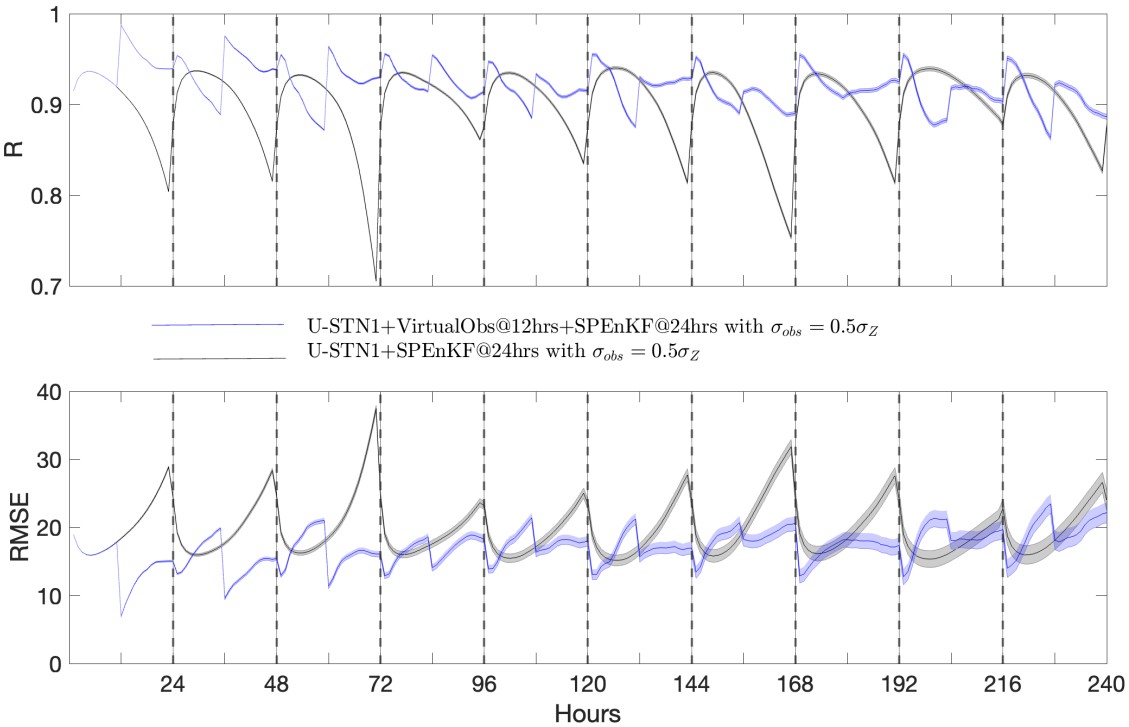

**Figure 8.** Performance of the the multi-time-step U-STNx+SPEnKF framework (with virtual observations at the 12<sup>th</sup> hour of every 24 h DA cycle) compared to that of the U-STN+SPEnKF framework for the case with $\sigma_{obs} = 0.5\sigma_Z$. The top (bottom) panel show R (RMSE in meters). The black, dashed vertical lines indicate DA of noisy observations at every 24 h. Forecasts are started from 30 random, noisy initial conditions. The lines (shading) show the mean (standard deviation) of the 30 forecasts.

## 5 Discussion and Summary

In this paper, we propose three novel components for DDWP frameworks to improve their performance. These components are: 1) a deep spatial transformer in the latent space to preserve equivariances and encode the relative spatial relationships of features of the spatio-temporal data in the network architecture, 2) a stable and inexpensive ensemble-based DA algorithm to ingest noisy observations and correct the forecast trajectory, and 3) a multi-time-step algorithm, in which the accurate forecasts of a DDWP model that uses a larger time step are assimilated as virtual observations into the less accurate forecasts of a DDWP that uses a smaller time step, thus improving the accuracy of forecasts at short intervals.

To show the benefits of each component, we use downsampled Z500 data from ERA5 reanalysis and examine the short-term forecast accuracy of the DDWP framework. To summarize the findings:

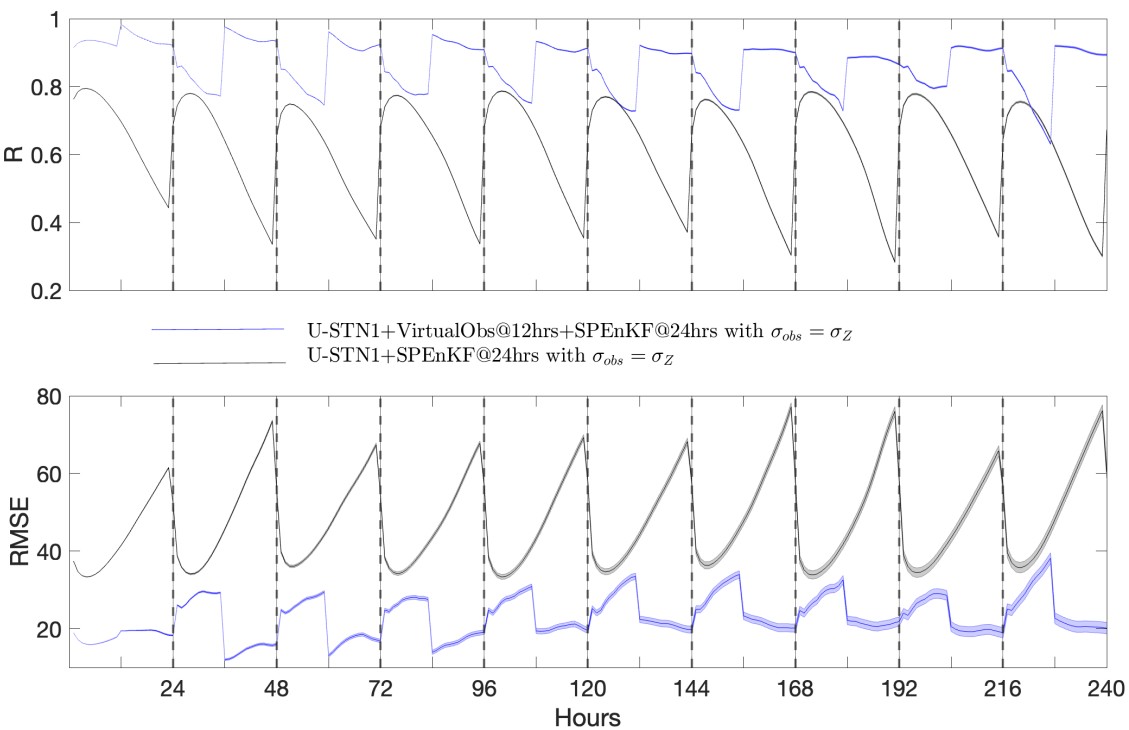

**Figure 9.** Same as Fig. 8 but with large observation noise, $\sigma_{\text{obs}} = \sigma_Z$.

1. As show in Section 4.1 for noise-free initial conditions (no DA), U-STN12, which uses a deep spatial transformer and $\Delta t = 12$ h, outperforms U-NET12, for example, extending the average prediction horizon (when ACC reaches 0.6) from 3.75 days (U-NET12) to 5.5 days (U-STN12). Examining a few examples of the spatio-temporal evolution of the forecasted Z500 patterns, we can see that U-STN better captures phenomena such as wavebreaking. We further show in Section 4.3 based on other metrics that with the same $\Delta t$, U-STN outperforms U-NET. These results demonstrate the benefits of adding deep spatial transforms to convolutional networks such as U-NETs.

2. As shown in Section 4.2, an SPEnKF DA algorithm is coupled with the U-STN1 model. In this framework, the U-STN1 serves as the forward model to data drivenly generate a large ensemble of forecasts in each DA cycle (24 h), when noisy observations are assimilated. Because U-STN1 is computationally inexpensive, for a state vector of size $D$, ensembles with $2D = 4096$ members are easily generated in each DA cycle, leading to stable, accurate forecasts without the need for localization or inflation of covariance matrices involved in the SPEnKF algorithm. The results show that DA can be readily coupled with DDWP models when dealing with noisy initial conditions. The results further show that such coupling is substantially facilitated by the fact that large ensembles can be easily generated with data-driven forward

models. Note however that NWP models have a larger number of state variables ($O(10^8)$) which would make SPEnKF very computationally expensive; in such cases, further parallelization of the SPEnKF algorithm would be required.

3. As shown in Section 4.3, the autoregressive DDWP models (U-STN or U-NET) are more accurate with larger $\Delta t$, which is attributed to the nonlinear error accumulation over time. Exploiting this trend and the ease of coupling DA with DDWP, we show that assimilating the forecasts of U-STN12 into U-STN1+SPEnKF as virtual observations in the middle of the 24 h DA cycles can substantially improve the performance of U-STN1+SPEnKF. These results demonstrate the benefits of the multi-time-step algorithm with virtual observations.

Note that to provide proof-of-concepts, here we have chosen specific parameters, approaches, and setups. However, the framework for adding these 3 components is extremely flexible, and other configurations can be easily accommodated. For example, other DA frequencies, $\Delta t$, U-NET architectures, or ensemble-based DA algorithms could be used. Furthermore, here we assume that the available observations are noisy but not sparse. The gain from adding DA to DDWP would be most significant when the observations are noisy and sparse. Moreover, the ability to generate $O(1000)$ ensembles inexpensively with a DDWP would be particularly beneficial for sparse observations for which the stability of DA is more difficult to achieve without localization and inflation (Asch et al., 2016). The advantages of the multi-time-step DDWP+DA framework would be most significant when multiple state variables, of different temporal scales, are used, or more importantly, when the DDWP model consists of several coupled data-driven models for different sets of state variables and processes (Reichstein et al., 2019; Schultz et al., 2021). Moreover, while here we show that ensemble-based DA algorithms can be inexpensively and stably coupled with DDWP models, variational DA algorithms (Bannister, 2017) could be also used, given that computing the adjoint for the DDWP models can be easily done using automatic differentiation.

The DDWP models are currently not as accurate as operational NWP models (Weyn et al., 2020; Arcomano et al., 2020; Rasp and Thuerey, 2021; Schultz et al., 2021). However, they can still be useful through generating large forecast ensembles (Weyn et al., 2021) and there is still much room for improving DDWP frameworks, for example using the three components introduced here as well as using transfer learning, which has been shown recently to work robustly and effectively across a range of problems (e.g., Ham et al., 2019; Chattopadhyay et al., 2020e; Subel et al., 2021; Guan et al., 2021).

Finally, we point out that while here we focus on weather forecasting, the three components can be readily adopted for other parts of the Earth system, such as ocean and land, for which there is a rapid growth of data and need for forecast and assimilation (e.g., Kumar et al., 2008b, a; Yin et al., 2011; Edwards et al., 2015; Liang et al., 2019).

# 6 Appendix

## 6.1 Forecast results with T850 Variable

In this section, we have shown an example of prediction performance with T850 instead of Z500. In Fig. 10, we can see that U-STN12 shows improved performance as compared to U-NET12 in T850 as well.

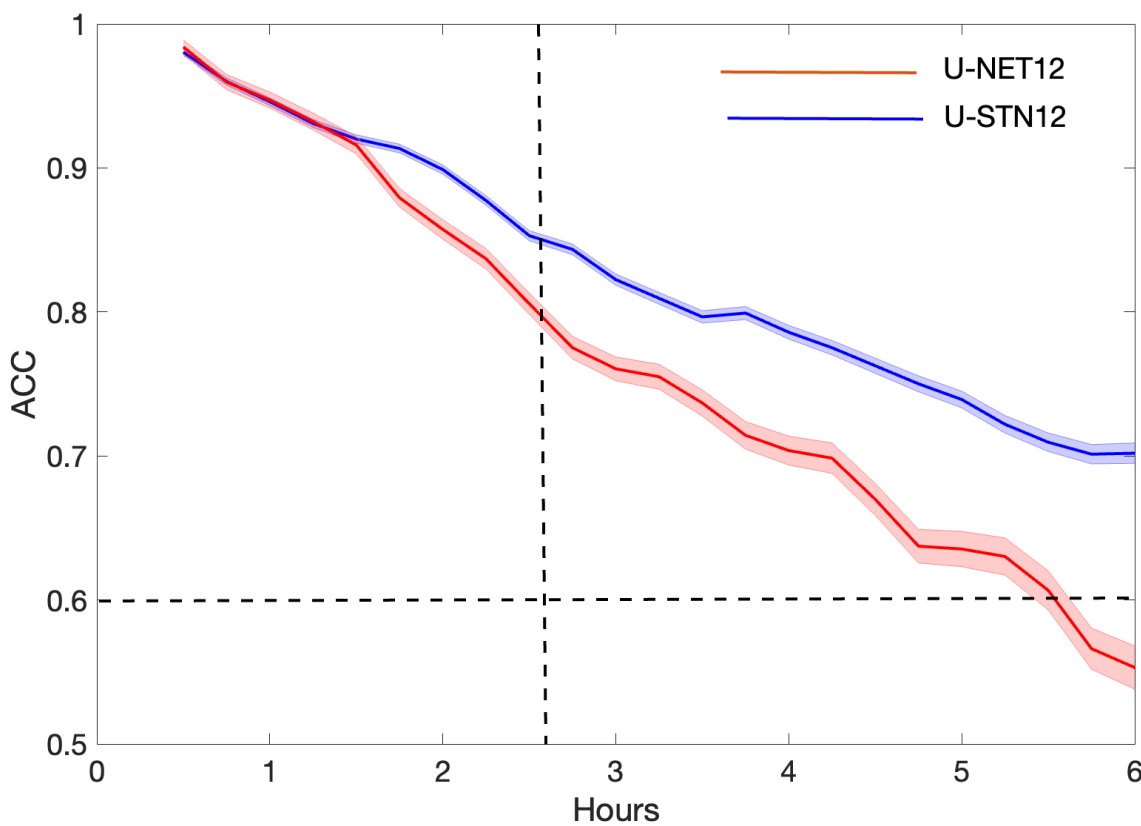

**Figure 10.** Performance of U-STN12 and U-NET12 on T850. Shading represents standard deviation over 30 initial conditions.

## 6.2 Comparison with two WeatherBench Models

In this section, we present Table 3 to compare U-STN12 model with two WeatherBench models at day 3 and day 5 in terms of RMSE ($m^2 s^{-2}$) for Z500.

| Models | RMSE ($m^2 s^{-2}$) at 3 days | RMSE ($m^2 s^{-2}$) at 5 days |
|:---:|:---:|:---:|
| Linear Regression (direct) from WeatherBench | 693 | 783 |
| CNN (direct) from WeatherBench | 626 | 757 |
| U-STN12 (our model) | 294 | 490 |

**Table 3.** Comparison of U-STN12 with two WeatherBench models

*Code and data availability.* All codes used in this study are publicly available at https://doi.org/10.5281/zenodo.5553570. The data are available from the WeatherBench repository at https://github.com/pangeo-data/WeatherBench.

*Author contributions.* A.C., M.M., and K.K. designed the study. A.C. conducted research. A.C. and P.H. wrote the manuscript. All authors
analyzed and discussed the results. All authors contributed to writing and editing of the manuscript.

*Competing interests.* The authors declare that they have no conflict of interest.

*Acknowledgements.* We thank Jaideep Pathak, Rambod Mojgani, and Ebrahim Nabizadeh for helpful discussions. We thank both the anonymous referees and the editor whose insightful comments, suggestions, and feedback have greatly improved the clarity of the manuscript. This work was started at National Energy Research Scientific Computing Center (NERSC) as a part of A.C.'s internship in the summer of 2020 under the mentorship of M.M. and K.K., and continued as a part of his PhD work at Rice University under the supervision of P.H. This research used resources of NERSC, a U.S. Department of Energy Office of Science User Facility operated under Contract No. DE-AC02-05CH11231. A.C. and P.H. were supported by ONR grant N00014-20-1-2722 and NASA grant 80NSSC17K0266. A.C. also thanks the Rice University Ken Kennedy Institute for a BP HPC Graduate Fellowship. E.B. was supported by the University of Maryland Flagship Fellowship and Ann G. Wylie Fellowship, and by Monsoon Mission II funding (Grant IITMMMIIUNIVMARYLANDUSA2018INT1) provided by the Ministry of Earth Science, Government of India.

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
