# Peer review of "Towards physically consistent data-driven weather forecasting: Integrating data assimilation with geometric deep learning in a case study with ERA5"

_Geoscientific Model Development, 2021_

## Referee Comment (RC2)

**Towards physically consistent data-driven weather forecasting: Integrating data assimilation with equivariance-preserving spatial transformers in a case study with ERA5**

The present manuscript introduces a new deep learning framework to forecast global geopotential height. More specifically, the authors introduce a U-NET (Sec 3.1, Fig 1) using circular convolutions (Sec 3.1.1) and augment it with an equivariance-preserving module (U-STN, Sec 3.1.2) to improve the overall accuracy of the forecast (Fig 3) and the consistency of the predicted patterns (Fig 4). They couple the resulting network with a sigma-point Ensemble Kalman filter (SPEnKF, Sec 3.2) that allows to assimilate "noisy observations" every 24 hours (Fig 5) or "virtual observations" every 12 hours produced by the same network run with a longer timestep (Fig 7). Evaluated using hourly, coarse-grained (Sec 2), ERA5 [2] meteorological reanalysis of 500-hPa geopotential height (Z500) from WeatherBench [3], the equivariance-preserving network using the SPEnKF to assimilate both "virtual" and "noisy observations" improves the performance of the same framework only assimilating "noisy observations" (Fig 8+9).

The manuscript is generally well-written, well-referenced, logically-structured; its figures are clear and its (surprisingly simple) code is accessible on GitHub (`https://github.com/ashesh6810/DDWP-DA`) and properly shared via Zenodo (`DOI10.5281/zenodo.4646676`). Given the methodological novelty and applicability of the U-STN+SPEnKF framework to data-driven weather forecasting, I recommend eventually publishing the present manuscript in *Geoscientific Model Development* (GMD). That being said, the article's impact may be hindered by incomplete benchmarking 1.1, a lack of testing on other meteorological variables 1.2, little justification of the equivariance-preserving module 1.3, and overly technical writing that may not be appropriate for GMD's audience 1.4. More details 1 and minor comments 2 are given below. I am optimistic that once improved, the manuscript will be a welcome addition to GMD and a helpful contribution to the sub-field of data-driven weather forecasting.

**Contents**

**1 Major Issues**

**1.1 Benchmarking**

[L220-244, Fig3] The manuscript's premise is that adding the equivariance-preserving module may improve the accuracy of data-driven weather forecasting, which is demonstrated by training a U-NET with and without the equivariance-preserving module and showing the resulting improvement in accuracy (as measured by the anomaly correlation coefficient) for lead times between 12 and 240 hours. This raises several issues:

- Despite using a well-defined benchmark (WeatherBench, [3]), the root mean squared error (RMSE) in is never calculated for the predictions without data assimilation (U-NET/U-STN), which prevents objective comparison with the baselines listed in Figure 2/Table 2 of [3]. I recommend at least calculating the RMSE in Z500 for lead times of 3 and 5 days to put the manuscript's results into the context of existing results (e.g., do U-NET/U-STN beat the simple linear regression leading to $\mathrm{RMSE}_{Z500}\,(3\mathrm{days}) \approx 693\mathrm{m}^2\mathrm{s}^{-2}$ **without data assimilation**? If the authors consider that only Z500 should be used as a predictor, then how do U-NET/U-STN perform compared to the linear regression equivalent that only uses Z500 as a predictor?).

- Once put into the WeatherBench context, it remains unclear whether U-STN systematically improves upon U-NET or if the result depends on the single set of (hyperparameters, weights, biases) explored in this manuscript. For instance, the only sensitivity explored in Figure 3 is that to initial conditions while the only sensitivity explored in Figure 6 is that to the timestep $\Delta t$ I recommend more thoroughly testing the addition of the equivariance-preserving module across:
  - Different weights and biases for a fixed set of hyperparameters by retraining U-NET/U-STN with different weights initializations and callbacks
  - Different hyperparameters by changing the convolutional and dense layers characteristics (number, width, kernel size) within the U-NET/U-STN architectures
  - Different architectures altogether: Would an equivariance-preserving module help an artificial neural network (with or without bottlenecks), simple linear models, etc.?
  In summary, I recommend conducting sensitivity tests to determine whether the paper's key conclusions hold across architectures, hyperparameters, and different weights/biases.

**1.2 Testing the Framework on Other Meteorological Variables**

[L105-110]

- Given that the manuscript's conclusions should apply to data-driven weather forecasting in general, I recommend testing the framework on a few more meteorological variables, especially given how easy it is to download variables from WeatherBench and how short the manuscript's repository code is. Natural choices would be variables benchmarked in WeatherBench, i.e. 850-hPa temperature (T850), 2m temperature (T2M) and total precipitation (TP).

- At the very least, the authors should discuss how appropriate equivariance-preserving spatial transformers are for thermodynamic variables like T850, which (in contrast to dynamic variables like Z500) directly respond to the strong planetary gradient in solar insolation. I recommend adding at least T850 to clarify the generality of the results e.g. presented in Figure 3.

**1.3 Justifying and Explaining Equivariance**

- [L130-140] In the other reference cited by the authors to justify using the spatial transformer module [4], the invariance under spatiotemporal translation, uniform motion, rotation/reflection, and scaling is justified for the Navier-Stokes and heat equation. However, when it comes to atmospheric dynamics, strong asymmetries exist in the horizontal (including but not limited to the Coriolis parameter for dynamical quantities, the solar insolation for thermodynamical quantity, and the land mass for all quantities). Therefore, I recommend carefully justifying why it would be appropriate to use an equivariance-preserving module in the text of subsection 3.1.2.

- [L233-236] Similarly, it would be helpful to more clearly justify/explain why equivariance-preserving networks would improve the representation of wave-breaking events, which are not rotationally nor translationally invariant. I

recommend more rigorously justifying that claim by e.g. zooming in Figure 4, adding more variables and network configurations, and not relying on "As discussed before" when the word "breaking" was simply listed in L120.

**1.4   Making the Manuscript more Accessible to GMD's Audience**

**1.4.1   Presentation of the Sigma-Point Ensemble Kalman Filter**

[L163-219] The authors adapt the Sigma-Point Ensemble Kalman Filter (SPEnKF, [1]) to augment their data-driven weather prediction framework with data assimilation (DDWP+DA). I found this description hard to follow because it lacks context and justification; I recommend revising the text to address the following questions:

- Do the authors strictly follow the derivations/methods of [1] or are there some key modifications to couple it to the ML prediction framework?

- Are analysis and "observations" used interchangeably here? In the affirmative, I would recommend sticking to one or the other.

- According to [1], SPEnKF is particularly well-suited for non Gaussian background/observation errors (e.g. multiplicative noise). Why then assume that $\epsilon$ be Gaussian, which (if I understand the derivation correctly) leads to Gaussian observational errors as $\boldsymbol{H} = \boldsymbol{I}$?

Additionally:
[L194] I recommend explicitly stating that representing observational noise using a random Gaussian process is a big approximation.
[L197] "with a certain level of uncertainty": I recommend explicitly stating that the uncertainty will be ideally represented by varying $\sigma_{\mathrm{obs}}$.
[L205] If $\boldsymbol{P_{ab}}$ is the cross-covariance matrix between the ensemble and observations, shouldn't the two last $Z_{\mathrm{ens}}$ be $Z_{\mathrm{obs}}$ in equation (8)?

**1.4.2   Overly technical vocabulary used throughout the manuscript**

GMD is targeted at the geosciences community: Although the community is relatively quite proficient in computational science, a lot of the vocabulary and technical terms used throughout the manuscript makes it difficult to read without ML background. To make the manuscript more accessible to the geoscientific community, I recommend:

- Quickly defining technical ML/DA terms used throughout the manuscript the first time they are introduced. This includes but is not limited to: convolutional neural network, deep spatial transformer, equivariance, Ensemble Kalman filter, encoding/decoding, autoregressive models, etc.

- Alternatively, adding a "ML definition" Table to the manuscript.

- Using more intuitive acronyms. For instance, U-STN, SPEnKF, and DDWP+DA are not particularly intuitive and may force the readers to go back and forth when reading the manuscript.

Also see comments in 2 to improve the manuscript's accessibility.

**1.5   Reproducibility**

**1.5.1   Unet's architecture**

[L113-L122] After checking the U-NET's architecture at `www.github.com/ashesh6810/DDWP-DA/blob/master/Unet_STN.ipynb` (same script as the one shared via Zenodo if I am not mistaken), I noticed that Figure 1 was not representative of the architectures used for U-NET/U-STN, which include additional dense layers after the convolutional layers. Additionally, because the authors do not disclose the type of pooling layers used in Figure 1, the architecture of the main algorithms used in the manuscript cannot be reproduced from the text.

- As the authors cite [5], I recommend following their Table 1 to transparently share the U-NET's architecture.

- Additionally, it would be nice to explicitly list the differences between [5] and this manuscript's U-NET, including (but not limited to) the presence of dense layers) to facilitate the comparison between the two frameworks.

**1.5.2   Weights and Biases**

[L113-122] The weights and biases of the neural networks are not shared (to my knowledge) in the code's repository, making the manuscript non reproducible. I highly encourage the authors to share the weights and biases of their networks for reproducibility purposes.

**1.5.3 Equivariance-preserving Module**

[L131-140] The authors do not provide enough details to help readers implement the spatial transformer module. I recommend explicitly stating how to implement this module (the Bilinear Interpolation Class at `https://github.com/ashesh6810/DDWP-DA/blob/master/layers.py`). This could be done by e.g.:

- adding an "algorithm" in the manuscript's text, and

- giving more context for why the spatial transformer module requires adding a bi-linear interpolation kernel between the convolutions and the up-sampling.

**2 Minor Comments**

[L37-39] Although it has been demonstrated for low-dimensional systems in fluid dynamics, it is not trivial that:

- Incorporating physical constraints into a physics-agnostic data-driven weather prediction framework would require less data and hence remedy the short training set problem,

- the equivariance-preserving spatial transformer introduced in this manuscript can be used to enforce physical constraints.

I recommend rephrasing these introductory sentences or carefully justifying these two claims.
[L97] Is "data-drivenly" correct?
[L94-99] These three points, especially the third one, are extremely technical and hard to understand without re-reading them several times. Would it be possible to rephrase them?
[L105-110] This section is extremely short: Would it be possible to

- add more context for why the authors first decided to test the framework on Z500 specifically,

- add the number of samples for each training set, and

- add a short justification for the training/validation/test split chosen by the authors?

[L154] becomes a U-NET $\rightarrow$ becomes a standard U-NET?
[L160] (Over the baseline, U-NET) $\rightarrow$ Benchmarking against another quick fit by the authors is far from rigorous: Following the major comment 1.1, would it be possible to add a subsection to Section 2 or at least a paragraph in Section 3.3 to describe and justify the paper's benchmarking methods?
[L164] "unscented transformation" requires more context for readers who are not versed in the Ensemble Kalman filter
[L177-178] ~50-100 members are used $\rightarrow$ Missing reference: Are the authors referring to the Integrated Forecasting System?
[Fig2 caption] "DA ... DDWP" $\rightarrow$ Consider spelling out acronyms or rephrasing to facilitate the caption's readability.
[L193] $k \in [-D, -D+1, ...D-1, D]$ $\rightarrow$ Do the authors mean $k \in \{-D, -D+1, ...D-1, D\}$ or equivalently $k \in [\![-D, D]\!]$?
[L262-264] I find this claim confusing:

- Is ERA5 truly noise-free?

- Doesn't the de-noising property come from the fact that U-STN is a deterministic neural network, which by definition cannot produce noise?

- Or are the authors referring to the fact that U-STN has a filtering effect that makes the normalized output variance smaller than the normalized input variance? If that is the case, I recommend clarifying the text and quantitatively justifying this claim about U-STN.

[L273-275] This qualitative explanation ignores the fact that errors made by the neural network are larger (in physical units) for larger $\Delta t$. Therefore, I recommend clarifying that the error accumulation is larger than the error increase with $\Delta t$. Would it be possible to quantitatively verify that claim (e.g. via a supplemental figure)?
[L337] "variational DA algorithms": Which variational DA algorithms are the authors referring to? Ideally, provide references for readers who are less familiar with DA.
[L347-348] Would it be possible to add the GitHub repository's link as it may be more convenient than downloading the archived code for some readers?

**References**

[1] J. T. Ambadan and Y. Tang. Sigma-point particle filter for parameter estimation in a multiplicative noise environment. *Journal of Advances in Modeling Earth Systems*, 3(4), apr 2011.

[2] H. Hersbach and H. The ERA5 Atmospheric Reanalysis. *American Geophysical Union, Fall General Assembly 2016, abstract id. NG33D-01*, 2016.

[3] S. Rasp, P. D. Dueben, S. Scher, J. A. Weyn, S. Mouatadid, and N. Thuerey. WeatherBench: A benchmark dataset for data-driven weather forecasting. feb 2020.

[4] R. Wang, R. Walters, and R. Yu. Incorporating Symmetry into Deep Dynamics Models for Improved Generalization. 2020.

[5] J. A. Weyn, D. R. Durran, and R. Caruana. Improving Data-Driven Global Weather Prediction Using Deep Convolutional Neural Networks on a Cubed Sphere. *Journal of Advances in Modeling Earth Systems*, 12(9):e2020MS002109, sep 2020.

---

## Author Comment (AC1)

**Referee 1**

*This manuscript explores improvements in the rapidly advancing field of data-driven weather prediction (DDWP). Broadly, DDWP seeks to train empirical weather-prediction models based on deep learning architectures, such as convolutional neural networks, that have proved very successful in fields such as image processing. This work fits within the WeatherBench forecasting challenge, which aims to forecast the global 500hPa geopotential height field, given the same field at an earlier time.*

*One of the leading approaches for DDWP is to use a convolutional U-NET architecture in which the first ("encoding") half projects the higher-resolution geopotential height field onto one or more lower-resolution "latent spaces" or "encoding spaces" and the second ("decoding") half of the U-NET upsamples the results, via many convolutional layers, to the original space. Broadly, the convolutional blocks are learning how to project geopotential height features forward in time, with the different levels of the U-NET allowing different scales to be projected using different convolutional blocks. The first advance is, at the lowest-level encoding space of the U-NET, to add an "equivariance preserving spatial transformer"; the resulting network is known as U-STN and improves forecast quality over the U-NET. The spatial transformer appears to permit additional capabilities for rotation, scaling and translation of the encoded geopotential height features within the empirical model, which are helpful for improving the forecast performance. The addition of the spatial transformer is justified as providing additional capabilities to preserve equivariance to important symmetries in the fluid dynamics of the atmosphere and therefore to provide a more physics-aware neural network. I would like to see more justification for this interpretation, and more precision in its discussion (see below).*

*The second advance in the manuscript is to couple a data assimilation algorithm to the DDWP model. Currently the Weather Bench framework provides high-quality gridded initial conditions from which to run DDWP forecasts, therefore missing a major step in the broader challenge of weather forecasting, which is to create those gridded initial conditions by assimilating the diverse and sparse (non-gridded) weather observations. To explore this side of the problem, noise is added to the gridded initial conditions, which are then assimilated every 24h using an ensemble data assimilation algorithm. An interesting aspect is to use the low-cost DDWP to create much larger ensembles (around 4000 members) than are possible in typical NWP (around 50 - 100 members). This allows a novel ensemble DA algorithm to be used (one that can be coded in a few lines of python), apparently without the problems of covariance localisation that are required with smaller ensembles. A second application of DA is also presented, where it is used to merge forecasts from DDWP models with different integration lengths.*

*This is novel and interesting work, which may have substantial impact on the development of DDWP, and hence it is worthy of eventual publication. However, there are a few major issues to consider beforehand, including the previously mentioned issues around the physical interpretation.*

**Authors' response:**

We thank the referee for their positive evaluation of our manuscript. Based on the referee's suggestion we have revised our manuscript in blue. The referee's insightful suggestions have sufficiently improved the clarity of the manuscript. Herein, we provide point-by-point responses to the referee's comments.

*Referee's comment:*

*1) The idea of equivariance is introduced precisely in Wang et al. (2020), for example, as applying to a function f(x) given a symmetry group g. The function is equivariant to g if the result of applying any of the symmetries (or transformations) from the group is the same whether applied to the functions inputs or outputs: f(g x) = g f(x). The same paper lists the symmetries of the Navier-Stokes equations as space and time translation, uniform motion, reflect/rotation and scaling. By contrast, the current manuscript is in places vague about what it means by equivariance, and it does not anywhere show whether it is preserved in the models presented. The presentation and analysis of the results relating to equivariance needs to be improved:*

*(i) Through the manuscript there are statements referring to the U-STN as "the equivariance-preserving DDWP introduced here" (line 118). However, the baseline U-NET is also likely to be equivariance-preserving, at least to translation and reflection. The improved U-STN may add equivariance to a certain set of symmetries (the authors suggest reflection, rotation and scaling). The point being that both the U-NET and the USTN are likely equivariance-preserving to some degree, but neither of them in a complete way to all possible symmetries. In terms of analysis, it would be important to more precisely specify or confirm which symmetries are preserved, or to acknowledge if the exact set of symmetries preserved is unknown. In terms of presentation, to describe the USTN as equivariance-preserving and to imply the U-NET is not could be misleading, and the title might also be changed to better reflect this.*

**Authors' response:**

We thank the referee for raising this interesting point about symmetry groups. In the manuscript, we have explained that equivariance-preserving networks do not impose *a priori* symmetry inside the networks and rather optimizes parameters to learn the symmetry. It is also true that U-STN does not learn all symmetries. In our U-STN, we have implemented rotational, translational, and scaling transformation through six parameters in the $T(\theta)$ matrix (defined in section 3.1.2) connected to the latent space of the network. It does not impose rotational invariance, i.e., it does not enforce the output to remain invariant to rotation in the input. The spatial transformer module learns the transformation (only on the encoded latent space) such that the latent space decodes to the correct output (which may have undergone rotational, translational, or scaling transformation) during training. The U-NET, like a regular CNN, is invariant to translation. In the U-STN, we have only performed rotational, translational, and scaling transformation on the latent space of the encoder which we have further clarified in the revised manuscript between Lines 150-152. We

have changed the title to remove the word equivariance-preserving (and spatial transformers) and have kept the word geometric deep learning. Equivariance is a topic of interest in the geometric deep learning community [1] and we have thus added a reference to a recent survey in geometric deep learning (Bronstein et al., 2021) in Table 1.

*Referee's comment:*

*(ii) This work adds an affine transformation and an interpolation (manuscript equations 1 and 2) in the latent space of the encoder; this is referred to as a "spatial transformer" and described as creating a new coordinate system, which is then passed to the decoding part of the U-NET. On line 138 - 139 it is said "The spatial transformer module ensures that the latent space that is encoded is equivariance-preserving". First, given the definition of equivariance, it is hard to see how a latent space could be equivariance-preserving. Rather, it would be the relevant function, i.e. the spatial transformer, that is equivariance preserving. In any case, this assertion needs to be properly backed up. As a concrete example, to be equivariance-preserving to rotations, it would need to be shown that all rotations of features in the encoding space (the input to the spatial transformer) would provide identical results to those performed in the transformed space acted on by the decoder (the output of the spatial transformer).*

**Authors' response:**

We thank the referee for their insightful comment. By saying that the "latent space" is equivariance preserving, we meant to say that the nonlinear function that operates on the latent space of the U-NET captures the transformation given by $T(\theta)$ between the input of the latent space and the output of the decoder and is clarified in Line 149 in the revised manuscript. However, given the complexity of the feature space in the Z500 field, it is difficult to interpret whether the transformation in the latent space of the U-NET leads to physically meaningful transformations in the decoded output. For a simpler dataset such as the rotating MNIST, spatial transformer networks have been shown to capture meaningful rotational features in Jaderberg et al., [2]. We show that, in data-driven forecasting of weather, the spatial transformation (in the latent space) allows us to obtain a better prediction horizon as compared to the U-NET (without such a transformation). However, we agree with the referee that such interpretations of the transformation in the latent space should be pursued further in future studies. We have revised our manuscript between Lines 262-265 to reflect the difficulty in interpreting the precise effect of the transformation induced by $T(\theta)$ in complex physical flows such as the large-scale circulation.

*Referee's comment:*

*(iii) An alternative explanation for the success of the U-STN would be to think of the spatial transformer as being able to learn a transformation that is helpful to propagating the encoding-space version of the geopotential height field forward in time. The spatial transformer is described by a single 2x3 transformation matrix, T(theta), with 6 trainable parameters (manuscript equation 2), followed by an interpolation This can only learn to perform one transformation, and for example, it might have learnt a particular combination of rotation and translation helpful to propagating the encoding space equivalents of Rossby waves forward in time. To better understand what is going on from a physical point of view, it would be really helpful if the authors could present the 6 parameters of T(theta) and try to interpret their effect in these terms: what does the learned transformation do (e.g rotation, scaling, translation?), does it make physical sense?*

**Authors' response:**

We thank the referee for their insightful comment. Firstly, we agree with the referee that $T(\theta)$ learns only one transformation between the latent space of the network and the decoded output which is a combination of rotation, translation, and scaling. It is difficult to precisely interpret how the transformation given by $T(\theta)$ in the latent space captures specific features in a complex flow field such as Z500. We have tried interpreting the parameters, $\theta$, but because $T(\theta)$ is only applied to the latent space, it is very difficult to perform any interpretations as to which features are captured in the complex Z500 field. We agree with the referee that with further development in geometric deep learning [1], such exercises in interpretation, especially in complex physical flows should be undertaken in the future. We believe however, that we should first start with simpler atmospheric models such as quasi-geostrophic flow where such analysis towards interpretation should be performed. We are currently working towards such interpretation through a hierarchy of simpler atmospheric/ocean models.

*Referee's comment:*

*2) The level of methodological detail in the manuscript is not fully sufficient to allow replication of the results or to communicate the approach at a sufficient level of detail. The neural networks being used are not fully described in the manuscript. A better example would be Weyn et al. (2020) who have shown how it is possible to properly document a complex network structure within a paper, such as by providing a table describing the layers, tensor sizes, etc.. It would also be helpful to have more details on the technical implementation such as the use of Python, Keras and Tensorflow, for example.*

**Authors' response:**

We thank the referee for their helpful suggestion. Following the referee's suggestion, we have added Table 2 in the revised manuscript where we have documented the detailed difference between U-NET and U-STN architecture and the framework in which they have been implemented.

*Referee's comment:*

*3) Some source code is provided on Xenodo, and it helped me a lot in understanding the work. However, it still left a lot unclear, and I believe it may only be a sample from all the code used by the authors while performing their work. For example, the training details appear to have been placed within a Jupyter notebook (Unet_STN.ipynb), but it is not fully clear whether this applies to all three examples in the manuscript, and to both the U-NET and the U-STN, and to the three different training time ranges (1,3 or 12 h), which is unlikely. The definition of the U-STN network in the Jupiter notebook is very different from the ones in the EnKF examples, which is confusing - see attached file "u_stn_diff.txt". It is not clear whether the U-net definition is provided at all. I would have expected a standardised definition of the networks in a separate file that could be used by all different configurations. Generally, the code package could be made more helpful to other people by better documentation and/or comments, better code structure and standardisation, and by the provision of some or all of the relevant data files - in particular the network weights of the U-NET and U-STN.*

**Authors' response:**

We thank the referee for going through our code in such details and providing helpful suggestions to improve the readability of our code. We have worked on organizing our source code more carefully and have now provided the networks' weights and biases. We have uploaded the same on Zenodo and updated our Github.

*Referee's comment:*

*Minor issues*
*1) Line 24: "...promising results with fully data-driven weather prediction (DDWP) models that are trained on variables representing the large-scale circulation obtained from numerical models or reanalysis products (Scher, 2018; Weyn et al., 2019, 2020; Chattopadhyay et al., 2020d, a; Rasp et al., 2020; Arcomano et al., 2020; Chantry et al., 2021; Grönquist et al., 2021; Watson-Parris, 2021; Scher and Messori, 2021)". Not all of the citations here are presenting the DDWP of the large-scale circulation - for example, Watson-Parris (2021) and Chantry et al. (2021) are opinion pieces and Grönquist et al. (2021) concerns postprocessing. This is a helpful bibliography and I am not suggesting the removal of any of the citations. Rather it might be worth giving a few more words to categorise these works more precisely. Further, this list is missing a key reference in Rasp and Thuerey (2020), which is discussed by the authors just afterwards.*

**Authors' response:**

We thank the referee for this helpful suggestion. We have revised Lines 25-31 to incorporate this suggestion.

*Referee's comment:*

*2) Line 30: "... DDWP models may not suffer from some of the biases of physics-based, operational numerical weather prediction (NWP) models ...". It seems unnecessarily restrictive to mention only bias here; the aim is to reduce model uncertainty in general.*

**Authors' response:**

We thank the referee for this helpful suggestion. We have revised Line 34 to incorporate this suggestion and referred to the "biases of physics-based, operational numerical weather prediction (NWP) models" as model errors in general.

Referee's comment:

*3) Line 40: "… to equip these DDWP models with data assimilation (DA) …". As written, the role of data assimilation is left uncertain. Although DA is introduced more fully later in the introduction, it could still be helpful to give slightly more clarity here, for example "to run these DDWP models within a data assimilation framework to provide the initial conditions for the forecasts".*

**Authors' response:**

We thank the referee for this helpful suggestion. We have revised Lines 46-47 in the revised manuscript to incorporate this suggestion.

*Referee's comment:*

*4) Line 68 gives the first mention of the U-NET architecture in the paper; a citation or two might be handy, and/or a pointer to the parts of the paper that describe what it is.*

**Authors' response:**

We thank the referee for this helpful suggestion. We have now added a reference to the original U-NET paper in Line 75 in the revised manuscript.

*Referee's comment:*

*5) Line 74: "DA algorithm that corrects the trajectory of the atmospheric states every 6 h with observations from remote sensing and in-situ measurements" - every 6h is overly restricitve, ERA5 for example is produced on a 12h cycle.*

**Authors' response:**

We thank the referee for pointing this out. We have revised the manuscript (Line 81) to say that 6 h is an example time interval at which DA is performed.

*Referee's comment:*

*6) Line 117 "The baseline DDWP model used here is a U-NET similar to the one used in Weyn et al. (2020)" - as in main point 2, I would have found it helpful to have more description of the baseline U-NET, and it would be nice to know more precisely what is different compared to Weyn et al.*

**Authors' response:**

We thank the referee for this helpful suggestion. We have revised the manuscript between Lines 179-184 to briefly talk about the difference between the architectures of the two U-NETs. However, we want to emphasize that we are not presenting a benchmark DDWP model that competes with the DDWP model in Weyn at al., [4]. In this manuscript, we are providing a proof-of-concept for a specific spatial transformation module in the latent space that may improve the performance of any DDWP architecture, and we have considered U-NET as an example.

*Referee's comment:*

*7) Line 118 mentions the deep spatial transformer in the method section for the first time; a citation to the original source would be helpful here, and also in section 3.1.2 where it is described in more detail.*

**Authors' response:**

We have added the reference to the original paper in Line 142.

*Referee's comment:*

*8) In the bibliography, the citation to Esteves et al. (2018) is mostly in lowercase.*

**Authors' response:**

Thank you. We have fixed the reference.

*Referee's comment:*

*9) Line 152 - 153: "All codes for these networks (as well as DA) have been made publicly*

*available on GitHub (see the Code Availability statement)." The codes are provided on Zenodo (not GitHub) but as described in main point 3, they do not appear to be complete.*

**Authors' response:**

The codes in the Github repository are complete but we have only uploaded the U-STN for one $\Delta t$ where that variable can be changed to incorporate any other $\Delta t$. We have now organized our source code so that it is readable and has more clarity. We thank the referee for taking the time to go through our codes and providing us with useful suggestions.

*Referee's comment:*

*10) Line 164 - please give a few words of explanation on the meaning of "unscented"*

**Authors' response:**

We have now added a reference to unscented transformations in Line 187.

*Referee's comment:*

*11) Line 184: in the DA algorithm, the singular vector decomposition of the analysis error covariance matrix is used to generate perturbations to create a new ensemble. However, in this work the ensemble is not propagated forward in time hour-by-hour, but is generated using the analysis error valid at "t" to represent the forecast error at "t+23dt", which is strictly incorrect. The forecast error at 23h is going to be much larger than the analysis error at 0h, therefore the ensemble created in the current work is most likely an underestimate of the spread of the background error. This needs to be discussed in the manuscript.*

**Authors' response:**

We thank the referee for pointing this out. Carrying the ensembles for 24 h is computationally expensive especially since the ensemble size is very large (4096). We agree that the ensemble spread is an underestimate of the background error in this work and have highlighted that in the revised manuscript in Lines 212-213. However, we have performed experiments by propagating the ensembles as well and have not found a significant difference in performance. This has also been reported in the revised manuscript in Lines 211-212.

*Referee's comment:*

*12) Equations 6 and 8 have identical right hand sides, but are labelled as different things (P_a and P_ab respectively). So something must be missing from the RHS to explain why they are different, or else P_a and P_ab are the same.*

**Authors' response:**

Thank you for pointing out this typo. We have fixed the equations in the revised manuscript.

*Referee's comment:*

*10) I found Figure 2 and 7 slightly confusing. The positioning of the states Z(t), Z(t+ dT) and so on below the U-STN1 blocks is confusing if the x-axis represents time; the small blue arrows are not helpful (suggesting that the states are external data coming into the process) and not consistently applied either. It could be more helpful to more clearly show the relation of the U-STN1 blocks to their inputs and outputs.*

**Authors' response:**

Thank you for this helpful suggestion. We have slightly changed the schematic to adjust the time stamps of the U-STN blocks.

*Referee's comment:*

*11) The forecast verification in Figure 3 is based on 30 random initial conditions (line 249). Seeing as it is so cheap to run the DDWP models, why not provide the results based on the full 2018 test period, to obtain more statistical significance? It is also odd to see the strong variability in skill, particularly in the U-STN12, from one verification time to the next. This might suggest that the verification is not as statistically significant as suggested by the standard deviation range provided. Verification of NWP forecasts is usually much smoother as a function of forecast range. Even for DDWP forecasts such as shown in Weyn et al. (2020, their figs 4 and 5) this usually seems to be the case.*

**Authors' response:**

We have conducted a Kolmogorov-Smirnov test between the difference of the mean ACC of U-STN12 and U-NET12. The difference is **not** statistically significant between 12 h and 36 h, it **is** statistically significant between 36 h and 240 h.

*Referee's comment:*

*12) Another point on the comparison of U-STN12 to U-NET12, it would be really helpful to establish the quality of the U-NET12 baseline - how competitive is it with other DDWP models?*

**Authors' response:**

We have revised the manuscript between Lines 275-280 to compare between other baselines such as CNN and linear regression. However, we would like to emphasize that, in this paper, we do not intend to present the best DDWP model but rather a proof-of-concept on the advantage of using an equivariance-preserving module in the latent space and a framework to integrate DA with DDWP. Moreover, herein we present Table. 1 which compares the quality of performance of U-STN12, U-NET12 and the CNN and linear regression model in WeatherBench [5].

**Table 1.** A comparison between U-STN12 and U-NET12 presented in this manuscript and the linear regression and CNN models from WeatherBench [5]. Here, we have used the RMSE values of direct linear regression and CNN prediction at day 3 and day 5. Here "direct" refers to the models trained to predict Z500 directly at 3 and 5 days instead of iteratively predicting every hour.

| Models | RMSE $(m^2 s^{-2})$ at 3 days | RMSE $(m^2 s^{-2})$ at 5 days |
|---|---|---|
| Linear regression (direct) from WeatherBench | 693 | 783 |
| CNN (direct) from WeatherBench | 626 | 757 |
| U-STN12 (our model) | **294** | **490** |
| U-NET12 (our baseline model) | **310** | **517** |

*Referee's comment:*

*13) Line 261-262: "The reason behind the further improvement of the performance after DA is the de-noising capability of neural networks (Xie et al., 2012)" - this seems overly confident given that it has not been demonstrated in the manuscript: "A likely reason behind the further improvement ..." would be a fairer description.*

**Authors' response:**

Thank you. We have revised the manuscript in Lines 298-300 based on the referee's suggestion.

*Referee's comment:*

*14) Section 4.3 gives an example of the use of DA to merge forecasts of different lengths. I find this section helpful as an illustration of the skill variations obtained with cycled (autoregressive) predictions versus direct predictions. However, instead of using DA, why not just throw away the cycled U-STN1 state at t+12dt and replace it by the forecast from U-STN12? It would be good to see if the DA can actually improve on that; in other words whether the cycled U-STN1 is bringing some additional information that is worth preserving.*

**Authors' response:**

We agree with referee's point. The directly predicted state at $t + 12\Delta t$ would probably be skillful as well. In fact, Liu et al., [3] had used interpolation instead of DA to obtained skillful autoregressive prediction with a multi-step framework. Further, not using DA would reduce the computational cost of the framework as well. However, the skill of the prediction by using the predicted state at $t + 12\Delta t$ from U-STN12 also depends on the value of initial noise as well. A fair comparison would require conducting further systematic experiments to see how the effect of noise affects recycling the state at $t + 12\Delta t$ in comparison to performing DA with it. In this paper, we show that performing DA with it is simply one way to recycle the obtained state and there may well be other methods such as interpolation (as shown in Liu et al., [3]) or simply using the state itself for further predictions that may yield skillfull predictions.

*Referee's comment:*

*16) Conclusions / discussion: on the benefits of DDWP for DA algorithms, item 2 line 314: being able to generate an ensemble large enough to provide fully-sampled background error covariance matrix is a major benefit here. However, the state vector size in the current work (2048) is still quite small compared to what might be expected in a more sophisticated DDWP approach, let alone NWP, where the state vector size is approaching 10^8. It should be acknowledged and discussed that the ability to use the SPEnKF algorithm, and to dispense with localisation, is not just the speed of the model (the DDWP) but the small size of the state vector.*

**Authors' response:**

Thank you. We have revised the text accordingly between Lines 356-357.

**References:**

1. Bronstein, M., Bruna, J., Cohen, T., and Velickovic, P., Geometric Deep Learning: Grids, Groups, Graphs, Geodesics, and Gauges, arXiv preprint arXiv:2104.13478,2021.

2. Jaderberg, M., Simonyan, K., Zisserman, A., et al.: Spatial transformer networks, in: Advances in Neural Information Processing Systems, pp. 2017–2025, 2015.

3. Liu, Y., Kutz, J. N., and Brunton, S. L., Hierarchical Deep Learning of Multiscale Differential Equation Time-Steppers, arXiv preprintarXiv:2008.09768, 2020.

4. Weyn, J.A., Durran, D.R., and Caruanna, R., Improving Data-Driven Global Weather Prediction Using Deep Convolutional Neural Networks on a Cubed Sphere, of Advances in Modeling Earth Systems, 12(9):e2020MS002109, 2020.

5. Rasp, S., Dueben, P. D., Scher, S., Weyn, J. A., Mouatadid, S., and Thuerey, N.: WeatherBench: A Benchmark Data Set for Data-Driven Weather Forecasting, Journal of Advances in Modeling Earth Systems, 12, e2020MS002 203, 2020.

---

## Author Comment (AC2)

**Referee 2**

*The present manuscript introduces a new deep learning framework to forecast global geopotential height. More specifically, the authors introduce a U-NET (Sec 3.1, Fig 1) using circular convolutions (Sec 3.1.1) and augment it with an equivariance-preserving module (U-STN, Sec 3.1.2) to improve the overall accuracy of the forecast (Fig 3) and the consistency of the predicted patterns (Fig 4). They couple the resulting network with a sigma-point Ensemble Kalman filter (SPEnKF, Sec 3.2) that allows to assimilate "noisy observations" every 24 hours (Fig 5) or "virtual observations" every 12 hours produced by the same network run with a longer timestep (Fig 7). Evaluated using hourly, coarse-grained (Sec 2), ERA5 [2] meteorological reanalysis of 500-hPa geopotential height (Z500) from WeatherBench [3], the equivariance preserving network using the SPEnKF to assimilate both "virtual" and "noisy observations" improves the performance of the same framework only assimilating "noisy observations" (Fig 8+9).*

*The manuscript is generally well-written, well-referenced, logically structured; its figures are clear and its (surprisingly simple) code is accessible on GitHub (https://github.com/ashesh6810/DDWP-DA) and properly shared via Zenodo (DOI10.5281/zenodo.4646676). Given the methodological novelty and applicability of the U-STN+SPEnKF framework to data-driven weather forecasting, I recommend eventually publishing the present manuscript in Geoscientific Model Development (GMD). That being said, the article's impact may be hindered by incomplete benchmarking 1.1, a lack of testing on other meteorological variables 1.2, little justification of the equivariance-preserving module 1.3, and overly technical writing that may not be appropriate for GMD's audience 1.4. More details 1 and minor comments 2 are given below. I am optimistic that once improved, the manuscript will be a welcome addition to GMD and a helpful contribution to the sub-field of data-driven weather forecasting.*

We thank the referee for their positive and constructive comments on the contribution of this manuscript to the field of data-driven weather forecasting. Their comments have helped us immensely in improving the clarity of the manuscript. All changes in the revised manuscript have been highlighted in blue. Herein, we present point-by-point responses to the referee's comments.

*Referee's comment:*
1 Major Issues
*1.1 Benchmarking*
*[L220-244, Fig3] The manuscript's premise is that adding the equivariance-preserving module may improve the accuracy of data-driven weather forecasting, which is demonstrated by training a U-NET with and without the equivariance-preserving module and showing the resulting improvement in accuracy (as measured by the anomaly correlation coefficient) for lead times between 12 and 240 hours. This raises several issues:*

- *Despite using a well-defined benchmark (WeatherBench, [3]), the root mean squared error (RMSE) in is never calculated for the predictions without data assimilation (U-NET/U-STN), which prevents objective comparison with the baselines listed in Figure 2/Table 2 of [3]. I recommend at least calculating the RMSE in Z500 for lead times of 3 and 5 days to put the manuscript's results into the context of existing results (e.g., do U-NET/U-STN beat the simple linear regression leading to $RMSE_{Z500}$ (3days) $693\ m^2 s^{-2}$ without data*

*assimilation? If the authors consider that only Z500 should be used as a predictor, then how do U-NET/U-STN perform compared to the linear regression equivalent that only uses Z500 as a predictor?).*

**Author's response:** We thank the referee for raising the question of comparing our data-driven model with the existing models in the WeatherBench [1] benchmark. We have, in our manuscript, reported the RMSE values of both U-STNx and U-NETx models (where x is 1, 6, and 12) in Figure 6 (left panel). We show that the best U-STNx model, i.e., U-STN12 have RMSE of $30m$ or $294 \, m^2 s^{-2}$ ($30m \times 9.8 ms^{-2}$ where $g = 9.8 \, ms^{-2}$ is the acceleration due to gravity) in Z500 at 3 days of lead time and $50m$ or $490 \, m^2 s^{-2}$ at 5 days of lead time without data assimilation (DA). Herein, Table 1, we present the comparison of U-STN12 with the models in WeatherBench [1] for lead time of 3 and 5 days from Figure 2 (of the WeatherBench [1] paper) as indicated by the referee.

**Table 1.** A comparison between U-STN12 presented in this manuscript and the linear regression and CNN models from WeatherBench [1]. Here, we have used the RMSE values of direct linear regression and CNN prediction at day 3 and day 5. Here "direct" refers to the models trained to predict Z500 directly at 3 and 5 days instead of iteratively predicting every hour.

| Models | RMSE $(m^2 s^{-2})$ at 3 days | RMSE $(m^2 s^{-2})$ at 5 days |
|---|---|---|
| Linear regression (direct) from WeatherBench | 693 | 783 |
| CNN (direct) from WeatherBench | 626 | 757 |
| U-STN12 (our model) | **294** | **490** |

As shown in Table 1, U-STN12 outperforms both linear regression and CNN models presented in WeatherBench. We have revised the manuscript to reflect this fact in Lines 275-280 in section 4.1. A short comparison between U-STN12 and the models in WeatherBench [1] has been provided in those above mentioned line numbers. However, we emphasize here, that the objective of our paper is not to present the best data-driven weather prediction (DDWP) model. Instead, we intend to show that a spatial-transformer module inside any DDWP architecture may improve the performance of the model and is shown by considering U-NET as an example architecture. While exploring all DDWP architectures is beyond the scope of this manuscript, other studies such as Wang et al., 2020 [2] has also shown the usefulness of equivariance-preserving architecture in spatio-temporal prediction of turbulent flow. Furthermore, we show in this paper, that a DDWP model can be integrated with DA without loss in stability of the DA algorithm or any indication of filter divergence which we have further explained in section 4.2. With such an integration of DDWP and DA, we also propose a proof-of-concept for a novel multi-step framework for improving the performance of the DDWP+DA model in section 4.3.

*Referee's comment:*

- *Once put into the WeatherBench context, it remains unclear whether U-STN systematically improves upon U-NET or if the result depends on the single set of (hyperparameters, weights, biases) explored in this manuscript. For instance, the only sensitivity explored in*

*Figure 3 is that to initial conditions while the only sensitivity explored in Figure 6 is that to the timestep Δt I recommend more thoroughly testing the addition of the equivariance-preserving module across:*

*    - Different weights and biases for a fixed set of hyperparameters by retraining U-NET/U-STN with different weights initializations and callbacks*

*    - Different hyperparameters by changing the convolutional and dense layers characteristics (number, width, kernel size) within the U-NET/U-STN architectures*

*    - Different architectures altogether: Would an equivariance-preserving module help an artificial neural network (with or without bottlenecks), simple linear models, etc.?*

*In summary, I recommend conducting sensitivity tests to determine whether the paper's key conclusions hold across architectures, hyperparameters, and different weights/biases.*

**Authors' response:**

We thank the referee for pointing out this very practical need for thorough hyperparameter optimization (HPO) when presenting the performance of a model. In this manuscript, we have performed HPO thoroughly through extensive trial and error. We have independently optimized the hyperparameters of U-NETx and U-STNx. Specifically, we have considered the effect of changing the:

- Weight initialization (e.g., Gaussian random, log-normal, and Xavier) and seen that the generalization error (RMSE during validation) of the architecture is not sensitive to the initialization.

- The size of the convolution kernel ($5 \times 5$), number of dense layers in the STN module (4), and the number of neurons in each of the 4 dense layers ($500, 200, 100, 50$) have been chosen after significant trial and error over these reported numbers.

- The equivariance-preserving module has been shown to be useful in convolutional architectures (regular encoder-decoder in Jaderberg et al., 2015 [3], U-NET in Wang et al., 2020 [2], and convolutional Res-Net in Wang et al., 2020 [2]). In most complex spatio-temporal modeling, 2D fields of states or observables are used to train the deep learning models. Such models are inherently convolutional in nature to account for the 2D fields on which they are trained. Fully-connected neural networks lose information about spatial correlation of the 2D fields and would perform poorly in predicting 2D fields. Hence, the advantage of equivariance inside such architectures may not be apparent. However, the theory of equivariance, as clearly explained in Wang et al, 2020 [2] and more recently in geometric deep learning by Bronstein et al., 2021 [3] is applicable to any architecture. More thorough analysis on the choice of architecture can be performed through very computationally expensive neural architectures search (NAS) as shown in Liu et al., 2018 [4]. An application of NAS in geophysical fluid dynamics has been shown in Maulik et al., 2020 [5] on the Argonne Leadership Computing Facility. However, owing to limited computational resources, NAS could not be performed in this study.

We have revised the manuscript in section 3 to clarify the extensive trail-and-error that has been performed on the hyperparameters of the architecture (Lines 175-178 and Caption of Table 2) and

have also cited the work on NAS for geophysical turbulence in Lines 160-162. We further emphasize here, that our objective is not to present the most performant deep learning architecture as the DDWP model, but to provide a proof-of-concept of the advantage of equivariance and then show the possibility of integration of DDWP with DA which builds into our novel multi-step framework for DDWP+DA.

*Referee's comment:*

*1.2 Testing the Framework on Other Meteorological Variables*

- *Given that the manuscript's conclusions should apply to data-driven weather forecasting in general, I recommend testing the framework on a few more meteorological variables, especially given how easy it is to download variables from WeatherBench and how short the manuscript's repository code is. Natural choices would be variables benchmarked in WeatherBench, i.e. 850-hPa temperature (T850), 2m temperature (T2M) and total precipitation (TP).*

- *At the very least, the authors should discuss how appropriate equivariance-preserving spatial transformers are for thermodynamic variables like T850, which (in contrast to dynamic variables like Z500) directly respond to the strong planetary gradient in solar insolation. I recommend adding at least T850 to clarify the generality of the results e.g. presented in Figure 3.*

**Authors' response:**

We thank the referee for pointing out this question about the generalizability of the architecture to predict on other meteorological variables. We agree with the referee that it is relatively easy to test the architecture on other variables. Following the referee's suggestion, we have conducted experiments to determine how well we can predict T850 with U-STN12 as compared to U-NET12. In Figure 1, shown here, we report the ACC of T850 with U-STN12 and U-NET12 (with U-STN12 outperforming U-NET12) which shows an improved prediction horizon as compared to Z500. This is likely due to the slow-moving nature of T850. We would like to clarify that in this experiment, we have used both Z500 and T850 as input to U-STN12 in 2 separate channels. We have revised the manuscript to add this information in the text in section 4.1 in Lines 271-274. However, in this manuscript, we intend to show a proof-of-concept wherein DA has been integrated with a DDWP model. In order to show that, we have taken Z500, simply as an example.

[Figure]

Figure 1. Anomaly correlation coefficient (ACC) calculated between T850 anomalies of ERA5 and T850 anomalies predicted using USTN12 and U-NET12 from 30 noise-free, random initial conditions. The solid lines and the shadings show the mean and the standard deviation over the 30 initial conditions.

As shown in Figure 1, U-STN12 performs well in terms of predicting T850 and outperforms U-NET12 as well. In general, as shown in multiple papers (cited inside the manuscript) including the WeatherBench paper, deep learning architectures can be used to predict on several meteorological variables and our finding is consistent with those studies.

*Referee's comment:*

*1.3 Justifying and Explaining Equivariance*

- *In the other reference cited by the authors to justify using the spatial transformer module [4], the invariance under spatiotemporal translation, uniform motion, rotation/reflection, and scaling is justified for the Navier-Stokes and heat equation. However, when it comes to atmospheric dynamics, strong asymmetries exist in the horizontal (including but not limited to the Coriolis parameter for dynamical quantities, the solar insolation for thermodynamical quantity, and the land mass for all quantities). Therefore, I recommend*

*carefully justifying why it would be appropriate to use an equivariance-preserving module in the text of subsection 3.1.2.*

**Authors' response:**

We thank the referee for raising this important question of whether rotation, reflection, or translational symmetries exist in atmospheric dynamics. We agree with the referee that it indeed does not have to be the case. In fact, equivariance is a property that accounts for the lack of symmetry in rotation and tracks the rotational features in the spatio-temporal flows. Conventional CNNs tend to enforce rotational symmetry while equivariance-preserving module ensures that the symmetry is not preserved. The affine transformation inside the module tries to learn the rotation and scaling of features as the input is passed through the U-STN. We have further added a more comprehensive and recent review in geometric deep learning by Brochstein et al., 2021 [3] that explains the theory of equivariance and its application in deep learning in the revised manuscript. In the revised manuscript, between Lines 63-65, we clearly explain how equivariance ensures that an *a priori* rotational symmetry is *not* imposed within the architecture (at least in the latent space, in our architecture). We have further highlighted this and justified the use of an equivariance-preserving module in section 3.1.2 between Lines 150-152 in the revised manuscript.

*Referee's comment:*

- *Similarly, it would be helpful to more clearly justify/explain why equivariance-preserving networks would improve the representation of wave-breaking events, which are not rotationally nor translationally invariant. I recommend more rigorously justifying that claim by e.g., zooming in Figure 4, adding more variables and network configurations, and not relying on "As discussed before" when the word "breaking" was simply listed in L120.*

**Authors' response:**

We thank the referee for their comment. As explained in the previous response, equivariance ensures that we do not *a priori* impose rotational symmetry in the deep learning architecture. It accounts for relative change in positions of features that comes from rotation, translation, and scaling. However, it is still unclear (and hard to prove) that wavebreaking events can be captured simply with an equivariance-preserving module since wavebreaking is a very nonlinear process. We simply speculate that an equivariance-preserving network *may* improve the overall performance of prediction so that it captures *some* wave-breaking events. We have revised the manuscript (between Lines 255-259) to reflect this speculation and do not suggest that the equivariance-preserving module is either necessary or sufficient to capture wavebreaking, and this may very well be just an example. We have removed all lines that may suggest that equivariance-preserving modules may lead to better representation of wavebreaking.

*Referee's comment:*

*1.4 Making the Manuscript more Accessible to GMD's Audience*

*1.4.1 Presentation of the Sigma-Point Ensemble Kalman Filter*

*The authors adapt the Sigma-Point Ensemble Kalman Filter (SPEnKF, [1]) to augment their data-driven weather prediction framework with data assimilation (DDWP+DA). I found this description hard to follow because it lacks context and justification; I recommend revising the text to address the following questions:*

- *Do the authors strictly follow the derivations/methods of [1] or are there some key modifications to couple it to the ML prediction framework?*

- *Are analysis and "observations" used interchangeably here? In the affirmative, I would recommend sticking to one or the other.*

- *According to [1], SPEnKF is particularly well-suited for non-Gaussian background/observation errors (e.g. multiplicative noise). Why then assume that $\epsilon$ be Gaussian, which (if I understand the derivation correctly) leads to Gaussian observational errors as $H = I$?*

*Additionally:*
*[L194] I recommend explicitly stating that representing observational noise using a random Gaussian process is a big approximation. [L197] "with a certain level of uncertainty": I recommend explicitly stating that the uncertainty will be ideally represented by varying $\sigma_{obs}$. [L205] If $P_{ab}$ is the cross-covariance matrix between the ensemble and observations, shouldn't the two last $Z_{ens}$ be $Z_{obs}$ in equation (8)?*

**Authors' response:**

We thank the referee for their helpful suggestion to improve the clarity of our presentation of SPEnKF in the manuscript.

- We have followed the method outlined in Ambadan et al., 2011 [6].

- No, analysis and observations are not used interchangeably. Analysis is obtained from Eq. (9) in section 3.2. Observations are generated from the ERA5 data by adding Gaussian noise with 0 mean and $\sigma_{obs}$ standard deviation. We have clarified this in the revised manuscript in Lines 233-234 in section 3.2.

- In this paper, we have considered a simple case where Gaussian observation noise is added to the true ERA5 data as is common in most DA literature [7,8,9]. This allows for a simple linear $H$ operator in the form of the identity matrix $I$. Indeed, SPEnKF can account for non-Gaussian observation noise. Several other types of DA techniques such as particle filters [10] can also be used for non-Gaussian observation noise. However, such techniques are computationally intractable with high-dimensional systems. A DDWP model for particle generation in particle filters can also enable application of particle filters in high-dimensional systems.

We have revised our manuscript (Line 233) to explain that Gaussian noise in observations is an approximation but is used widely in literature and that the uncertainty in the observed state is given by $\sigma_{obs}$ in Lines 218-219.

- We had a typo in the equations. We have now fixed that in the revised manuscript.

*Referee's comment:*

*1.4.2 Overly technical vocabulary used throughout the manuscript*

*GMD is targeted at the geosciences community: Although the community is relatively quite proficient in computational science, a lot of the vocabulary and technical terms used throughout the manuscript makes it difficult to read without ML background. To make the manuscript more accessible to the geoscientific community, I recommend:*

- *Quickly defining technical ML/DA terms used throughout the manuscript the first time they are introduced. This includes but is not limited to: convolutional neural network, deep spatial transformer, equivariance, Ensemble Kalman filter, encoding/decoding, autoregressive models, etc.*

- *Alternatively, adding a "ML definition" Table to the manuscript.*

- *Using more intuitive acronyms. For instance, U-STN, SPEnKF, and DDWP+DA are not particularly intuitive and may force the readers to go back and forth when reading the manuscript.*

*Also see comments in 2 to improve the manuscript's accessibility.*

**Authors' response:**

We thank the referee immensely for their helpful suggestions to improve the clarity and accessibility of the manuscript to the geosciences audience. Considering the suggestions, we have added a table in the revised manuscript, Table 1, where we have defined (with short descriptions) all ML/DA related, and framework related acronyms suggested by the referee. We hope that the inclusion of this table would improve the clarity of this manuscript further to the geosciences audience.

*Referee comment*

*1.5 Reproducibility*

*1.5.1 Unet's architecture*

*[L113-L122] After checking the U-NET's architecture at www.github.com/ashesh6810/DDWP-DA/blob/master/Unet_STN.ipynb (same script as the one shared via Zenodo if I am not mistaken), I noticed that Figure 1 was not representative of the architectures used for U-NET/U-STN, which include additional dense layers after the convolutional layers. Additionally, because the authors do not disclose the type of pooling layers used in Figure 1, the architecture of the main algorithms used in the manuscript cannot be reproduced from the text.*

- *As the authors cite [5], I recommend following their Table 1 to transparently share the U-NET's architecture.*
- *Additionally, it would be nice to explicitly list the differences between [5] and this manuscript's U-NET, including (but not limited to) the presence of dense layers) to facilitate the comparison between the two frameworks.*

**Authors' response:**

We thank the referee for their helpful suggestion on improving the reproducibility of the architecture used in this paper. We have slightly revised Figure 1 to include the dense layers which are a part of the localization network. The figure is used only as a schematic. We have further added Table 2 in the revised manuscript to include the detailed information on the exact architecture of U-STNx and U-NETx.

- We have further revised the manuscript between Lines 179-184 to highlight the difference between the architecture used in Weyn et al., [11] and ours. Here, we would like to emphasize that we do not claim that our architecture is more performant as compared to that used in Weyn et al., [11], but is simply the one we have chosen based on extensive trail-and-error in terms of HPO. Moreover, the architecture presented in Weyn et al., [11] uses data on a cubed sphere rather than a rectangular grid so is distinctly different from the framework shown in this paper.

*Referee's comment:*

*1.5.2 Weights and Biases*

*The weights and biases of the neural networks are not shared (to my knowledge) in the code's repository, making the manuscript non reproducible. I highly encourage the authors to share the weights and biases of their networks for reproducibility purposes.*

**Authors' response:**
Thank you for this excellent suggestion. We have now shared the weights and biases HDF5 file for reproducibility purposes.

*Referee's comment:*

*1.5.3 Equivariance-preserving Module*

*The authors do not provide enough details to help readers implement the spatial transformer module. I recommend explicitly stating how to implement this module (the Bilinear Interpolation Class at https://github.com/ ashesh6810/DDWP-DA/blob/master/layers.py). This could be done by e.g.:*
- *adding an "algorithm" in the manuscript's text, and*
- *giving more context for why the spatial transformer module requires adding a bi-linear interpolation kernel between the convolutions and the up-sampling.*

**Authors' response:**

We thank the referee for their helpful suggestion. We have cited the original paper that had introduced STNs, Jaderberg et al., [12] in Line 142 in section 3.1.2 where details about the need for a differentiable interpolation kernel (bilinear interpolation in this case) has been clearly explained. The justification for using the interpolation kernel is rather elaborate and we feel that such methodological details in the manuscript may distract the readers from a geoscience community from the main points of the paper, which is to introduce STN as an equivariance-preserving module and integrate DDWP models with DA algorithms for weather forecasting. We have also provided the code for implementing the bilinear interpolation layer for reproducibility.

*Referee's comment*

*2 Minor Comments*

*[L37-39] Although it has been demonstrated for low-dimensional systems in fluid dynamics, it is not trivial that:*

- *Incorporating physical constraints into a physics-agnostic data-driven weather prediction framework would require less data and hence remedy the short training set problem,*
- *the equivariance-preserving spatial transformer introduced in this manuscript can be used to enforce physical constraints.*

**Authors' response:**

We thank the referee for their insightful suggestions.

- We agree with the referee that previous literature in climate dynamics have not shown that physical constraints may help training neural networks with less training samples. However, a few recent papers have shown that physics-informed neural networks can be used to train on the shallow-water equations [13] and on high-dimensional fluid dynamics systems [14] with short training sets. It is thus promising to use physical constraints inside the neural architectures. We have addressed this in the revised manuscript and have revised Lines 42-44 accordingly.

- Preserving equivariance is not analogous to physical constraints in the network. The equivariance-preserving module *may* lead to better representation of rotational, translational, and scaling features in the architecture and thus lead to more physically consistent predictions. However, there is no guarantee that it would do so all the time. In the example shown in this paper, we report improved overall accuracy with the spatial-transformer module. As described in Kashinath et al. [15], we mention in the manuscript that it may be only "*one*" of the many ways to improve physical consistency in the neural architecture (Line 58 in the revised manuscript).

*Referee's comment:*

*I recommend rephrasing these introductory sentences or carefully justifying these two claims.*
*[L97] Is "data-drivenly" correct?*
*[L94-99] These three points, especially the third one, are extremely technical and hard to understand without re-reading*
*them several times. Would it be possible to rephrase them?*
*[L105-110] This section is extremely short: Would it be possible to*
- *add more context for why the authors first decided to test the framework on Z500 specifically,*
- *add the number of samples for each training set, and*
- *add a short justification for the training/validation/test split chosen by the authors?*

**Authors' response:**

We have changed "data-drivenly" to "data-driven" fashion in Line 104. We have slightly re-phrased point 3 to make it more clear.

- Since Z500 is representative of the large-scale dynamics in the troposphere, responsible for influencing near-surface weather, and extremes, we had decided to use Z500 as an example. As shown here (based on the referee's suggestion) we can also get equally good prediction performance for T850. Z500 has also been used in Rasp et al., 2020 [1], and Weyn et al., 2020 [11]. We have edited the revised manuscript (Lines 113-115) to explain our choice of using Z500 as the variable in this study. We have also revised the third point in Lines 105-106 based on the referee's suggestion.

- In the revised manuscript between Lines 117-118, we have explained that training data was obtained from years 1979-2015 ($\sim$315360 samples), validation data was obtained between 2016-2017 (17520 samples), and we tested on data from 2018 (8760 samples).

- We had not randomly sorted the entire ERA5 data in order to avoid correlation between training and testing sets. Hence, we had split the training and validation in the fashion described above and in Lines 117-118 in the manuscript.

*Referee's comment:*

*[L154] becomes a U-NET →becomes a standard U-NET?*

**Authors' response:**

Yes. We have revised Line 163 so that it says "standard U-NET"

*Referee's comment:*

*[L160] (Over the baseline, U-NET) Benchmarking against another quick fit by the authors is far from rigorous: Following the major comment 1.1, would it be possible to add a subsection to Section 2 or at least a paragraph in Section 3.3 to describe and justify the paper's benchmarking methods?*

**Authors' response:**

As suggested by the referee, we have shown in this response (Table 1) how U-STN12 compares against the benchmarks in the WeatherBench paper [1]. However, we emphasize that the paper is not presenting U-STN12 as a state-of-the-art DDWP for WeatherBench. In fact, we are only showing that an equivariance-preserving module instead an architecture *may* improve its prediction performance. Beyond that, we describe the integration of DA with DDWP models in section 4.2 and a novel multi-step framework in improve the performance of the DDWP+DA model in section 4.3. We have revised the manuscript in section 4.1 (Lines 275-280) to compare with the performance of linear regression and CNN from the WeatherBench paper [1]. However, since we are not presenting a benchmark for WeatherBench, we would prefer not to put a separate section on benchmarking with the WeatherBench models.

*Referee's comment:*

*[L164] "unscented transformation" requires more context for readers who are not versed in the Ensemble Kalman filter*

**Authors' response:**

We have added a reference to unscented transformations in ensemble Kalman filter in Line 187 in the revised manuscript.

*Referee's comment:*

*[L177-178] ~50-100 members are used. Missing reference: Are the authors referring to the Integrated Forecasting System?*

**Authors' response:**

Yes, we are. We have now added a reference to Line 201 in the revised manuscript.

*Referee's comment:*

*[Fig2 caption] "DA ... DDWP" → Consider spelling out acronyms or rephrasing to facilitate the caption's readability.*

**Authors' response:**

We thank the referee for pointing this out. We have now added Table 1 in the revised manuscript that explains the acronyms of the paper. We feel that it would make the captions too long and hard to read by spelling out the acronyms in the caption.

*Referee's comment:*

$k \in [-D, -D + 1, \ldots D - 1, D] \rightarrow$ Do the authors mean $k \in \{-D, -D + 1, \ldots D - 1, D\}$ or equivalently $k \in [[-D, D]]$?

**Authors' response:**

Yes, we are. We have changed the text accordingly in Line 217 in the revised manuscript. Thank you.

*Referee's comment:*

[L262-264] I find this claim confusing:
- Is ERA5 truly noise-free?
- Doesn't the de-noising property come from the fact that U-STN is a deterministic neural network, which by definition
cannot produce noise?
- Or are the authors referring to the fact that U-STN has a filtering effect that makes the normalized output variance smaller than the normalized input variance? If that is the case, I recommend clarifying the text and quantitatively justifying this claim about U-STN.

**Authors' response:**

We thank the referee for these interesting questions.

- Since ERA5 is obtained after data assimilation, we assume that it is noise free. In this study, since ERA5 is considered as the truth we have further added noise to ERA5 to

mimic observations. Generally, DA algorithms are presented with twin experiments, where the observations are obtained from adding noise to the truth and the analysis states are compared to the truth.

- Yes. Since the U-NET or U-STN is trained on non-noisy data, it is expected to not have any ability to represent noise in the output. We have revised the text in the revised manuscript between Lines 298-300 to reflect this.

*Referee's comment:*

*[L273-275] This qualitative explanation ignores the fact that errors made by the neural network are larger (in physical units) for larger Δt. Therefore, I recommend clarifying that the error accumulation is larger than the error increase with Δt.. Would it be possible to quantitatively verify that claim (e.g. via a supplemental figure)?*

**Authors' response:**

We thank the referee for this question. Note, all three DDWP models with $\Delta t = 1\,h$, $\Delta t = 6\,h$, and $\Delta t = 12\,h$ have the same generalization error which is close to 0.003 (for normalized Z500; normalization involves removing the mean and dividing by standard deviation of Z500 of the training set) in one time step of prediction. Therefore, the errors are not larger in physical units. In fact, this is what makes autoregressive prediction with larger $\Delta t$ more accurate as compared to iteratively predicting with a smaller $\Delta t$. However, there is an optimal $\Delta t$ after which generalization error would keep increasing with an increase in $\Delta t$. However, there is no apparent theoretical understanding as to what this optimal $\Delta t$ depends on for a chaotic system, e.g., neural architecture, system's dynamics, etc. We are currently working towards a theoretical understanding to the non-trivial dependence of error accumulation and error propagation through deep neural architectures for autoregressive prediction. However, at this point of time it is rather difficult to show concrete quantitative evidence on how error propagates through data-driven autoregressive models.

*Referee's comment:*

*"variational DA algorithms": Which variational DA algorithms are the authors referring to? Ideally, provide references for readers who are less familiar with DA.*

**Authors' response:**

We thank the referee for pointing this out. By variational DA algorithms, we mean 3D-Var and 4D-Var. We have added a reference to the same in Line 373 in the revised manuscript.

*Referee's comment:*

*[L347-348] Would it be possible to add the GitHub repository's link as it may be more convenient than downloading the archived code for some readers?*

**Author's response:**

We thank the referee for this helpful suggestion. We have added the Github repository in the code and data availability statement in Line 384 in the revised manuscript.

**References:**

1. Rasp, S., Dueben, P. D., Scher, S., Weyn, J. A., Mouatadid, S., and Thuerey, N.: WeatherBench: A Benchmark Data Set for Data-Driven Weather Forecasting, Journal of Advances in Modeling Earth Systems, 12, e2020MS002 203, 2020.

2. Wang, R., Walters, R., and Yu, R.: Incorporating Symmetry into Deep Dynamics Models for Improved Generalization, arXiv preprint arXiv:2002.03061, 2020.

3. Bronstein, M., Bruna, J., Cohen, T., and Velickovic, P., Geometric Deep Learning: Grids, Groups, Graphs, Geodesics, and Gauges, arXiv preprint arXiv:2104.13478,2021.

4. Liu, C., Zoph, B., Neumann, M., Shlens, J., et al., Progressive neural architecture search, In Proceedings of the European Conference on Computer Vision (EECV), 2018.

5. Maulik, R., Egele, R., Lusch, B., and Balaprakashan, P., Recurrent neural network architecture search for geophysical emulation, International Conference on High Performance Computing, Networking, Storage and Analysis, 2020.

6. Ambadan, J.T., and Tang, Y., Sigma-point particle filter for parameter estimation in a multiplicative noise environment. Journal of Advances in Modeling Earth Systems, 3(4), 2011.

7. Brajard, J., Carrassi, A., Bocquet, M., and Bertino, L., Combining data assimilation and machine learning to emulate a dynamical model from sparse and noisy observations: a case study with the Lorenz 96 model, Journal of Computational Science, 44, 101 171, 2020.

8.  Brajard, J., Carrassi, A., Bocquet, M., and Bertino, L., Combining data assimilation and machine learning to infer unresolved scale parametrization, Philosophical Transactions of the Royal Society A, 379, 20200 086, 2021.

9.  Carrassi, A., Bocquet, M., Bertino, L., and Evensen, G., Data assimilation in the geosciences: An overview of methods, issues, and perspectives, Wiley Interdisciplinary Reviews: Climate Change, 9, e535, 2018.

10. Paul, F., Kunsch, H.R., Particle filters and data assimilation, Annual Review of Statistics and its Application, 5, 421-449, 2018.

11. Weyn, J.A., Durran, D.R., and Caruanna, R., Improving Data-Driven Global Weather Prediction Using Deep Convolutional Neural Networks on a Cubed Sphere, of Advances in Modeling Earth Systems, 12(9):e2020MS002109, 2020.

12. Jaderberg, M., Simonyan, K., Zisserman, A., et al.: Spatial transformer networks, in: Advances in Neural Information Processing Systems, pp. 2017–2025, 2015

13. Bihlo, A., Popovych, R.O., Physics-informed neural networks for the shallow-water equations on the sphere, arxiv preprint arXiv:2104.00615

14. Raissi, M., Yazdani, A., Karniadakis, G., Hidden fluid mechanics: Learning velocity and pressure fields from flow visualizations, Science, 367, 1026-1030, 2020.

15. Kashinath, K., Mustafa, M., Albert, A., Wu, J., Jiang, C., Esmaeilzadeh, S., Azizzadenesheli, K., Wang, R., Chattopadhyay, A., Singh, A.,et al.: Physics-informed machine learning: case studies for weather and climate modelling, Philosophical Transactions of the Royal SocietyA, 379, 20200 093, 2021.

---

## Referee Report (RR1)

**Towards physically consistent data-driven weather forecasting: Integrating data assimilation with geometric deep learning in a case study with ERA5**

**First Round of Revisions**

Thank you for thoroughly responding to the reviewers' comments: I believe the manuscript is significantly clearer and more accessible as a result. Now that the manuscript is easier to read, it is apparent that U-STN may not be equivariance-preserving (1.1) and hence physically constrained (1.2), which requires major revisions (1) as equivariance preservation is a key point of the manuscript. Making this manuscript as scientifically rigorous as possible is even more important now that its preprint is already cited four times. Minor comments are listed in Section 2; I would recommend being particularly cautious as some of the minor revisions that were discussed by the authors in the response did not make it to the "tracked changes" version that we received from GMD. Even if U-STN were not equivariance-preserving, the framework presented by the authors still improves the accuracy of U-Net and would be useful to gain insight into the meteorological prediction problem at hand (1.3), and I still recommend this manuscript for eventual publication in GMD once the network's equivariance properties are clarified.

**Contents**

**1 Major Issues**

**1.1 U-STN may not be equivariance-preserving**

**1.1.1 Inconsistencies**

It is now clear that the manuscript does not accurately describe the code that was used by the authors. Below are some inconsistencies that I have noticed:

- [L147] "The parameters $\theta$ are learned through back-propagation": This could mislead readers into thinking that the 6 parameters of $\boldsymbol{\theta}$ are uniquely learned through back-propagation, resulting in a single transformation (scaling+rotation+translation) enforced at evaluation time (after the network is trained). If I am not mistaken, this would **not** result in an equivariance-preserving network.

- Additionally, when looking at the code, I noticed that instead, $\boldsymbol{\theta}$ are not trainable parameters but rather outputs of the last dense layer before up-sampling. Therefore, there is one matrix $\boldsymbol{\theta}$ per sample, which is why the "transformations" tensor has the shape (Number of samples, 6) in the code of the (bilinear interpolation+affine transformation) layer.

- [L153-155] Without further clarification, I fail to see how the above framework preserves $SO(3)$ equivariance, which if I am not mistaken would here be defined along the lines of:

$$\forall \boldsymbol{\theta}, \forall \mathbf{Inputs}, \ \mathrm{U} - \mathrm{STN} \left[ \mathcal{T}_{\boldsymbol{\theta}} \left( \mathbf{Inputs} \right) \right] = \mathcal{T}_{\boldsymbol{\theta}} \left( \mathrm{U} - \mathrm{STN} \left[ \mathbf{Inputs} \right] \right).$$

  Using the authors' encoding framework, wouldn't this be closer to enforcing robustness of U-STN's outputs to a range of pre-determined $\boldsymbol{\theta}$ parameters regardless of the inputs, instead of learning the (scaling+rotation+translation) that maximizes accuracy for each sample separately?

- [Figure 1] When looking at this schematic, it looks like the localization network output is transformed by $\mathcal{T}_{\boldsymbol{\theta}}$, resulting in the circled cross output that is then fed to the $5 \times 5$ convolutional kernel. Looking at the current version of the code, it looks like instead, the localization network output is used to produce a set of transformations $\mathcal{T}_{(\mathrm{Sample}, \boldsymbol{\theta})}$, which is **then applied to the bilinearly interpolated version of the original input** $Z(t)$, and not the input to the latent space as written in [L152].

I recommend addressing each one of these 4 inconsistencies separately, by either clarifying the manuscript or correcting its code.

**1.1.2 Confirming the superiority of U-STN over U-Net**

The above inconsistencies made me wonder what exactly explains the accuracy gains of U-STN compared to its corresponding baseline U-Net. More specifically, to confirm that this superiority is indeed linked to the presence of the spatial transformer module, would it be possible to:

1. Transparently communicate the number of learnable weights/biases/parameters in U-Net and U-STN? From the code, it looks like U-STN has $2,461,965$ learnable parameters but I could not find the equivalent number for U-Net.

2. Additionally disclose the bottleneck size for U-Net and U-STN?

This would help affirm that U-STN performs better than U-Net because of the spatial transformer module and not simply because it has more learnable parameters or a larger bottleneck.

**1.2 As a result, U-STN may not be physically constrained**

[L43-45] U-STN is specifically introduced as a method to physically constrain a deep learning framework. Even if the authors use nuanced language, I find this motivation misleading for the reader, especially as both examples mentioned by the authors enforce invariance (which is different from equivariance, and even more different from the setup introduced in this manuscript): [2] weakly enforces invariants associated to the Navier-Stokes equations via the loss function, while [1] enforces the PDE structure (associated to invariants) and initial conditions as soft constraints and the boundary conditions as hard constraints in the case of the shallow water equations on a sphere. This is fundamentally different from

the more data-driven and flexible approach adopted by the authors in this manuscript: Arguably, no physical constraints are enforced on the outputs of U-STN.

Would it be possible to revise the introduction to clarify that this framework is not directly analogous to standard physics-informed approaches that can be found in the literature?

Note that these revisions would not necessarily decrease the manuscript's impact but simply clarify the exact motivation (e.g., interpretability) and benefits (e.g., improved accuracy) of this novel architecture.

**1.3 However, the current version of U-STN can help gain physical insight**

While U-STN does not seem to enforce physical constraints, its strength could rely on the low-dimensional, interpretable construction of $\mathcal{T}_{\theta}$. Once the authors confirm the superiority of U-STN over U-Net (1.1.2), it would be interesting to better understand how the transformation $\mathcal{T}_{\theta}$ associated to each sample improves the accuracy of the data-driven forecast. This could be done quickly and within the manuscript's scope by uniquely decomposing each transformation $\mathcal{T}_{\theta}$ into its corresponding scaling, rotation, and translation (e.g., using this decomposition `https://stackoverflow.com/questions/45159314/decompose-2d-transformation-matrix`, or any well-defined decomposition that the authors find interpretable). Once the transformation is decomposed, the authors could answer questions such as:

- How does the transformation vary from sample to sample (e.g., is the scaling, the rotation, or both different)?

- How does the transformation vary from field to field (e.g., does the transformation significantly change when adding temperature as an output)?

- What does this transformation mean physically (e.g., can the rotation or scaling be traced back to atmospheric wave properties)?

- Why does this transformation improve the accuracy of the data-driven forecast (e.g., are these just tunable parameters or is this transformation preventing erroneous assumptions about the rotational symmetry of e.g. wave breaking events as mentioned by the authors).

This could be a good way to motivate the introduction of the spatial transformer module if it cannot be clearly proven that it preserves equivariance.

**2 Minor Comments**

[Table 1] (Very minor) Would "change appropriately in response to a transformation" be clearer than "change appropriately to a transformation" here?

[Figure 1] Would it be possible to specify the bottleneck's size on this schematic, i.e. the shape of the localization network output?

[L198] Consider replacing $\mathcal{R}$ (\cal R) with $\mathbb{R}$ (\mathbb{R}) to follow conventions.

[Equation 6] This looks like an auto-correlation matrix rather than a covariance matrix: Is this equation correct?

[L272] "not shown for brevity": As the manuscript is already rich in ideas and technical details, I understand the authors' motivation to only focus on one field in the main text. However, since the manuscript significantly gains in generality (from a meteorological perspective) by showing the improved prediction

[Figures 5,6,8,9] Missing units for RMSE (it should be meters if I am not mistaken).

[Caption of Figure 5] The authors use parentheses for clarification, abbreviations, references, and I find that using it to express opposites is confusing in this caption (see [4] for a general discussion on the topic). More specifically, it looks like the coefficient of determination R is used to abbreviate RMSE. Would it be possible to rewrite the caption to avoid potentially confusing the reader?

[Figure 6, L275-280] Since it is difficult for readers to visualize numbers that are written in text, I highly recommend adding the WeatherBench baselines (and maybe [5, 3]) as scattered crosses on the left plot of Figure 6. This would facilitate the comparison between the performance of these baselines and U-STN, which would further underline the good performance of U-STN.

[L320] Typo: "At 24th hours".

[L382] "forecast/assimilation": Do the authors mean "forecast and assimilation"? The current phrasing may confuse readers.

[Code availability statement] To clarify, I was referring to the authors' GitHub repository (and not the WeatherBench repository). Would it be possible to share the GitHub of this manuscript specifically (corresponding to the Zenodo URL `https://zenodo.org/record/5553570#.YX-QKp5KhPY`) as part of the code availability statement?

**References**

[1] A. Bihlo and R. O. Popovych. Physics-informed neural networks for the shallow-water equations on the sphere. apr 2021.

[2] M. Raissi, A. Yazdani, and G. E. Karniadakis. Hidden fluid mechanics: Learning velocity and pressure fields from flow visualizations. *Science*, 367(6481):1026–1030, feb 2020.

[3] S. Rasp and N. Thuerey. Data-Driven Medium-Range Weather Prediction With a Resnet Pretrained on Climate Simulations: A New Model for WeatherBench. *Journal of Advances in Modeling Earth Systems*, 13(2):e2020MS002405, feb 2021.

[4] A. Robock. Parentheses are (are not) for references and clarification (saving space), 2010.

[5] J. A. Weyn, D. R. Durran, and R. Caruana. Improving Data-Driven Global Weather Prediction Using Deep Convolutional Neural Networks on a Cubed Sphere. *Journal of Advances in Modeling Earth Systems*, 12(9):e2020MS002109, sep 2020.

---

## Referee Report (RR2)

**Towards physically consistent data-driven weather forecasting: Integrating data assimilation with geometric deep learning in a case study with ERA5**

**Third Round of Revisions**

Thank you again for thoroughly responding to the reviewers' comments: At this point, the manuscript is clear and the authors' framework reproducible. However, the manuscript's first key point remains to be proven because (1) U-NETx and U-STNx are not compared in a systematic fashion (1.1); and (2) U-STN is still misleadingly presented as equivariance-preserving in parts of the manuscript (1.2). Additionally, WeatherBench models may be unfairly compared to a version of U-STN including data assimilation cycles (1.3), and U-STN is still framed as a physics-informed machine learning (ML) framework (1.4). Finally, there may still be discrepancies between the shared code and what is reported in the manuscript (2.1). I continue to think that this manuscript will be a welcome contribution to GMD once revised so as to not mislead readers about equivariance preservation. Overall, it would be helpful to clarify whether the improved test performance of U-STNx reported by the authors without data assimilation (e.g., Fig 6) is truly the result of some form of equivariance preservation, or whether using a bottleneck and a spatial transformer simply prevents overfitting the WeatherBench training set.

**Contents**

**1 Major Issues**

**1.1 Comparison between U-NET and U-STN**

The first key point of the manuscript relies comparing U-STN with U-NET to establish U-STN's improved performance compared to U-NET ("improving the forecast skill by 45%"). This requires being as precise as possible when comparing U-STN and U-NET:

- According to Table 2, U-STN has three additional layers (10/11/12) of 200/100/50 neurons. Would it be possible to transparently list the number of trainable parameters (e.g., the output of model.summary()) for both the U-STN and U-NET? If they are different, what would suggest that this is a fair comparison?

- For example, how is it possible to know whether U-STN improved performance over the test set comes from the spatial transformer or from the bottleneck created by layers 10/11/12. Would it be possible to introduce a U-NET with the exact same architecture as U-STN (minus the spatial transformer) to test whether the improved performance simply comes from not overfitting the WeatherBench training set?

- "the optimal set of hyperparameters that have been obtained after extensive trial and error": This is a vague statement in the age of formal hyperparameter optimization (e.g., Hyperas, Hyperopt, the Keras tuner, SHERPA, etc.) that does not establish in any way that the architecture of U-STN is comparable to that of U-NET. Would it be possible to:

- List the hyperparameters and their considered range during the manual search?
- Systematically report the performance of U-NET and U-STN over the training/validation/test set (as opposed to just the test set) to identify whether U-STN is a better fit, whether it generalizes better, etc.?
- This could also help better understand the role of the spatial transformer (e.g., does it help because it prevents overfitting the training set?) as equivariance is not generally preserved by the U-STN framework.

**1.2 U-STN is not equivariance-preserving**

U-STN is still framed as a framework preserving equivariance:

[L4 in the abstract] "to preserve a property called equivariance"

[L44] "based on building physical properties called equivariance"

[L102] "Introducing the equivariance-preserving"

[L338] "to preserve equivariances".

While the authors added a discussion at the end of subsection 3.1.2, the above statements would mislead most readers (especially those focusing on the abstract/introduction/conclusion) into thinking that this framework ensures equivariance preservation. Assuming that the above claims relate to $SO(3)$ equivariance, which if I am not mistaken would here be defined along the lines of:

$$\forall \boldsymbol{\theta}, \forall \mathbf{Inputs}, \ \mathrm{U-STN}\left[\mathcal{T}_{\boldsymbol{\theta}}\left(\mathbf{Inputs}\right)\right] = \mathcal{T}_{\boldsymbol{\theta}}\left(\mathrm{U-STN}\left[\mathbf{Inputs}\right]\right),$$

would it be possible to clarify that:

- The framework is not equivariant because it learns one set of $\boldsymbol{\theta}$ parameters per sample, and that these parameters are not physically interpretable according to the authors (in their response to reviews),

- The U-STN architecture includes **skip-connections**, which prevent equivariance preservation at multiple levels of the architecture (up6 and up7 in the code use conv2 and conv1).

Note that other manuscripts using equivariance preservation (e.g., [1]) typically define equivariance mathematically and exactly stipulate whether the property is satisfied or not upfront.

**1.3 Misleading comparison with WeatherBench**

[Table3, Sec 6.2]

1. Does the U-STN12 compared to WeatherBench models assimilate data from the test set? In the affirmative, I recommend comparing WeatherBench models to the version of U-STN12 that does not use data assimilation (e.g., the version presented in Fig 6) and clarifying the text accordingly.

2. Are the authors using the same test set (their paper uses 2018) as the test set used to report the WeatherBench models' performance (2017 and 2018 according to [2])? This would also prevent a fair comparison and I recommend using the same test set as in WeatherBench while explicitly acknowledging that samples from the validation set will be used for this particular comparison.

**1.4 U-STN is still framed as a physics-informed ML framework**

U-STN is still framed as a framework to improve physical consistency:

[Title] Towards physically-consistent

[L3 in the abstract] improve their physical consistency

[L44] provide a framework for [L41-42] incorporating physical constraints into the often physics-agnostic ML models

However, given that U-STN does not preserve equivariance (see 1.2), it is unclear why U-STN would make the predictions more physically consistent (other than the qualitative description in Sec 4.1). If it is impossible to quantitatively establish the improved physical consistency of U-STN predictions, I suggest removing the claims listed above.

**2 Minor Comments**

**2.1 Possible discrepancies between the code and the manuscript**

In the accompanying code for the baseline U-NETx, it seems that the authors are still training a U-STN ("stn()") for most of the training files instead of a U-NET ("unet_baseline()"). Would it be possible to explain this apparent discrepancy, which appears in all of the U-NETx ($x = $ 1hr, 12hr, etc.) scripts?

```
model.compile(loss='mse', optimizer='adam')
model.summary()
batch_size = 10      ### This has undergone HPO. Don't change
num_epochs = 8       #### This has undergone HPO. But less sensitive to change
lead = 1             #### See paper for details on this variable. This lead refers to "x" in U-NETx
count=0
for loop in fileList_train:
print('******************** counter*************',count)
File=nc.Dataset(loop)
Z=np.asarray(File['z'])
trainN=np.size(Z,0)-300
Z=(Z-M)/sdev
x_train=Z[0:trainN,:,:]
x_train=x_train.reshape([np.size(x_train,0),32,64,1])
y_train=Z[lead:trainN+lead,:,:]
y_train=y_train.reshape([np.size(y_train,0),32,64,1])
x_val= Z[trainN+lead:np.size(Z,0)-lead,:,:]
x_val=x_val.reshape([np.size(x_val,0),32,64,1])
y_val= Z[trainN+lead*2:np.size(Z,0),:,:]
y_val=y_val.reshape([np.size(y_val,0),32,64,1])
if (count>0):
 model = stn()
 model.compile(loss='mse', optimizer='adam')
 model.load_weights('best_weights_lead1.h5')
 hist = model.fit(x_train, y_train,
                 batch_size = batch_size,
         verbose=1,
         epochs = 20,
         validation_data=(x_val,y_val),shuffle=True,
         callbacks=[keras.callbacks.EarlyStopping(monitor='val_loss',
                             min_delta=0,
                             patience=5, # just to make sure we use a lot of patience before stopping
                             verbose=0, mode='auto'),
               keras.callbacks.ModelCheckpoint('best_weights_lead'+str(lead)+'.h5', monitor='val_loss',
                                 verbose=1, save_best_only=True,
                                 save_weights_only=True, mode='auto', period=1),history]
     )
else:
```

Figure 1: The U-NET ("Unet_noSTN") scripts appear to use the stn() model as soon as the model is trained on the second file of the training set.
From https://github.com/ashesh6810/DDWP-DA/blob/master/Unet_noSTN_lead1.py

**2.2   Typos**

[L26-27] "e.g." should be moved to the beginning of the parentheses
[L131] "indicates the $\Delta t$ that is used": Should e.g., "in units hours" be added to specify the units of $\Delta t$ in the U-STNx acronym?
[L352] "data-drivenly": Is this grammatically correct or should "in a data-driven fashion" be used instead here?
[L387] "have shown" → "show"

**References**

[1] C. Esteves, C. Allen-Blanchette, A. Makadia, and K. Daniilidis. Learning SO(3) Equivariant Representations with Spherical CNNs. *International Journal of Computer Vision*, 128(3):588–600, 2020.

[2] S. Rasp, P. D. Dueben, S. Scher, J. A. Weyn, S. Mouatadid, and N. Thuerey. WeatherBench: A benchmark dataset for data-driven weather forecasting. feb 2020.

---

## Author Response (AR2)

**Editor's Comments**

*Thank you for submitting reply in the Interactive Discussion, the authors have made the manuscript more consistent with the referee's arguments. In my role as editor of GMD, you will see that both referees had judged the manuscript to "reconsidered after major revisions".*

*First, please bring to your attention to the following requirement should meet in the Interactive Public Peer Review:*
*https://gmd.copernicus.org/articles/12/2215/2019/*
*https://www.geoscientific-model-development.net/about/manuscript_types.html*
*Please make sure that manuscript more Accessible to GMD's Audience. (see major Issues on original submission by reviewer 2).*

*Second, all codes and data for these networks should be publicly available on a persistent archive (GitHub does not). The codes of the manuscript are provided on Zenodo but they do not appear to be complete. Generally, the code package could be made more helpful to other people by better code structure and standardization, and by the provision of some or all of the relevant data files - in particular the network parameters of the U-NET and U-STN.*

*So, my decision at this stage is to send your final revised manuscript to referees for a further review. Good luck.*

**Authors' Response**

We thank the editor for their helpful comment. We have now edited and commented our code and sufficiently changed the structure to reflect modern standardization of machine learning codes while not sacrificing readability and understanding of the code structure. We have provided separate training files for each of our proposed models and baselines. We have uploaded the weights and biases files to facilitate reproducibility. We have put our codes on Zenodo and removed the link to Github (although Referee 2 had explicitly asked for it) since Github is not a persistent archive. However, in the comments, Referee 2 had mentioned that they have access to the codes on Github as well. We have updated those codes on Github as well.

---

## Author Response (AR3)

**Referee 1.**

**Referee comment**: Most of my comments have been addressed by a significant number of changes through the manuscript, and in the improved set of code archived on Zenodo. However, there is still need for improved presentation of the relevant neural networks, in order to make sure that the work can be correctly interpreted and reproduced. I put these as "major": although they are not extensive, there is one potentially important question over the fundamental methodological description.

*Authors' response:* We thank the referee for their positive assessment of our manuscript. Herein, we provide a point-by-point response to the referee's comments. All changes in the manuscript have been marked in blue. We also thank the referee for their helpful suggestions that have greatly improved the clarity of the manuscript.

**Referee comment:** The addition of Table 1 is a big step forward in documenting the relevant neural networks used in this study. However, comparing it to the code archived on Zenodo brings a number of questions.
- It would be useful to document the 2x2 resolution of the max pooling layers. This can be inferred from the text, but it would be useful to make that explicit.
- It would be useful to document input tensor shape/size for each layer in the table - this is most useful for layers 9-12 where it has not been clearly documented in the text.
- To properly understand the U-net architecture it is necessary to document the pass-throughs, e.g. the process of concatenating the output of one of the earlier higher-resolution layers with the output of an upsampling layer. For example, based on the function unet_baseline in Unet_noSTN_lead1.py, the upsampling layer 15 (as numbered in Table 1) is concatenated with the output of layer 5. This is illustrated in Figure 1, but it is helpful to have it in Table 1 as well.

*Authors' response:* We thank the referee for their insightful comments. We have now documented the $2 \times 2$ pooling layers in Table 2. We have also added a column to explicitly mention that output tensor shapes in Table 2. We have also explicitly mentioned the concatenation layers and which layer it has been concatenated with in Table 2. All these changes have been marked with blue.

**Referee comment:** Much of the text in section 3 seems to describe a single latent space at 8x16 resolution, but the U-NET also has one at 16x32 resolution.

*Authors' response:* The latent space for both U-STN and U-NET is $8 \times 16$. Two pooling layers would take the original sized input from $32 \times 64$ to $8 \times 16$. However, a latent space size of $16 \times 32$ would have little to no impact on the performance of both U-NET and U-STN.

**Referee comment:** Based on comparing the function Unet_noSTN_lead1.py and Unet_STN_lead1.ipynb, layers 9-12 in table 1 are only present in the Unet-STN: a note "Only for STN" needs to be added as for the layers 13 and 14.

*Authors' response:* We thank the referee for pointing this out. The note has been added in Table 2.

**Referee comment:** I may have misread the code, but the code does not seem to agree with the description of applying the spatial transformer to the lowest-resolution (8x16) latent space. The application of the spatial transformer is given in Unet_STN_lead1.ipynb by
x = BilinearInterpolation(sampling_size)([inputs, locnet]) In other words, the inputs are "locnet", which is the 6 elements of the affine transformation, generated by layers 9-12 (as described in Table 1) and "inputs" which is actually the input to the very first convolutional layer, in other words the geopotential height at the 32x64 resolution. Therefore the spatial transformer seems to be applied to the full-resolution data, and not in the latent space.

*Authors' response:* The indices for bilinear interpolation has been obtained from the original full-sized input but the transformation has been applied to the latent space as done in the original paper by Jaderberg et al., [1].

**Referee comment:** Figure 1 seems to show the spatial transformer being applied to the latent space at 16x32 resolution, not the one at 8x16 resolution as described in the text, or the original 32x64 space, as I have inferred from the code.

*Authors' response:* The two pooling layers reduce the latent space size to $8 \times 16$ from $32 \times 64$. The transformation is applied to the latent space after it goes through the dense layers as shown in Figure 1. Not all dense layers have been shown in the figure for compactness but reported in Table 2.

**Referee comment:** A couple of other points:

- My request for Figure 3 to be based on a larger sample of initial conditions than 30 was not addressed in the authors' response; there seems no reason not to include as many initial conditions as possible, in order to improve the comprehensiveness of the testing, even if the authors are confident that their statistical significance levels are well estimated.

*Authors' response:* We thank the referee for raising this important point of uncertainty quantification of the free prediction's ACC with both U-STN and U-NET. Due to limitations in computational resources, we were not able to perform predictions with more than 30 initial conditions. However, we have systematically performed predictions with 10 and 15 initial conditions and repeated the statistical test. We have found the significance testing robust across 10, 15, and 30 initial conditions. This shows that the error bounds are robust as well in Figure 3.

**Referee comment: -** Line 296: "stable beyond 10 days". In Figure 5 there is a visible trend in the standard deviation (the shaded area) across the 30 initial conditions, not yet discussed in the text. This suggests that the data assimilation is not fully stable even within the 10 days.

*Authors' response:* We thank the referee for asking this question about the stability of the SPEnKF+U-STN1 framework. As can be seen in Figure 5, for both levels of noise, both the RMSE

(R) curves does not visibly shoot up (down) monotonically with increasing DA cycles. At every DA cycle the R curves go up and RMSE curves go down to roughly the same value at each DA cycle. The uncertainty in these curves is derived from the spread in initial conditions and not the spread of the ensembles of the analysis state. We have (not shown for brevity in the text) also investigated the background covariance matrix and it shows localized correlation structure indicating that the SPEnKF filter is not diverging up to 10 days.

**References**:

1. Jaderberg, M., Simonyan, K., Zisserman, A., et al.: Spatial transformer networks, in: Advances in Neural Information Processing Systems, pp. 2017–2025, 2015.

**Referee 2**

Thank you for thoroughly responding to the reviewers' comments: I believe the manuscript is significantly clearer and more accessible as a result. Now that the manuscript is easier to read, it is apparent that U-STN may not be equivariance-preserving (1.1) and hence physically constrained (1.2), which requires major revisions (1) as equivariance preservation is a key point of the manuscript. Making this manuscript as scientifically rigorous as possible is even more important now that its preprint is already cited four times. Minor comments are listed in Section 2; I would recommend being particularly cautious as some of the minor revisions that were discussed by the authors in the response did not make it to the "tracked changes" version that we received from GMD. Even if U-STN were not equivariance-preserving, the framework presented by the authors still improves the accuracy of U-Net and would be useful to gain insight into the meteorological prediction problem at hand (1.3), and I still recommend this manuscript for eventual publication in GMD once the network's equivariance properties are clarified.

*Authors' response:* We thank the referee for their positive evaluation of our manuscript, insightful comments, and their support in making this manuscript more accessible. Herein, we provide a point-by-point response to the referee's comments. The changes in the manuscript have been tracked in blue. Following the referee's suggestions, we have also added an appendix section in the revised manuscript where we have reported the forecasting performance of T850 and a comparison with two WeatherBench models.

**Referee comments:**
**1 Major Issues**
**1.1 U-STN may not be equivariance-preserving**
**1.1.1 Inconsistencies**
It is now clear that the manuscript does not accurately describe the code that was used by the authors. Below are some inconsistencies that I have noticed:
[L147] "The parameters   are learned through back-propagation": This could mislead readers into thinking that the 6 parameters of   are uniquely learned through back-propagation, resulting in a single transformation (scaling+rotation+translation) enforced at evaluation time (after the network is trained). If I am not mistaken, this would not result in an equivariance-preserving network.

*Authors' response:* We agree with referee. We have revised the manuscript accordingly in Line 150 where we explicitly mention that the transformation between the input and output through the matrix $T(\theta)$ is equivariant and **not** the entire network. We have also revised Line 148 to indicate that the parameters $\theta$ are predicted for each sample.

Additionally, when looking at the code, I noticed that instead, are not trainable parameters but rather outputs of the last dense layer before up-sampling. Therefore, there is one matrix   per sample, which is why the "transformations" tensor has the shape (Number of samples; 6) in the code of the (bilinear interpolation+affine transformation) layer.

[L153-155] Without further clarification, I fail to see how the above framework preserves SO (3) equivariance, which if I am not mistaken would here be defined along the lines of:

$$\forall\,\theta, \forall Inputs, U - STN[T_\theta(inputs) = \tau_\theta(U - STN(inputs))$$

Using the authors' encoding framework, wouldn't this be closer to enforcing robustness of U-STN's outputs to a range of pre-determined parameters regardless of the inputs, instead of learning the (scaling+rotation+translation) that maximizes accuracy for each sample separately?

*Authors' response:* We have explicitly mentioned in the revised manuscript that the entire network is not equivariant by construction in Line 154. Only the transformation in the latent space performed through $T(\theta)$ is used to capture translation, rotation, and scaling. We have also mentioned in Line 148 that the parameters $\theta$ are predicted for each sample.

[Figure 1] When looking at this schematic, it looks like the localization network output is transformed by T, resulting in the circled cross output that is then fed to the 5x5 convolutional kernel. Looking at the current version of the code, it looks like instead, the localization network output is used to produce a set of transformations T (Sample;), which is then applied to the bilinearly interpolated version of the original input Z(t), and not the input to the latent space as written in [L152]. I recommend addressing each one of these 4 inconsistencies separately, by either clarifying the manuscript or correcting its code.

*Authors' response:* The referee is correct in pointing out the bilinear interpolation requires the original input $Z(t)$. However, that is used to calculate the indices of the interpolation kernel. It is however then used on the latent space connected to the dense layers, similar to the original implementation of the spatial transformer network in Jaderberg et al., [1].

**Referee's comment:**

**1.1.2 Confirming the superiority of U-STN over U-Net**
The above inconsistencies made me wonder what exactly explains the accuracy gains of U-STN compared to its corresponding baseline U-Net. More specifically, to confirm that this superiority is indeed linked to the presence of the spatial
transformer module, would it be possible to:
1. Transparently communicate the number of learnable weights/biases/parameters in U-Net and U-STN? From the code, it looks like U-STN has 2; 461; 965 learnable parameters but I could not find the equivalent number for U-Net.
2. Additionally disclose the bottleneck size for U-Net and U-STN?
This would help affirm that U-STN performs better than U-Net because of the spatial transformer module and not simply because it has more learnable parameters or a larger bottleneck.

*Authors' response:* This is an interesting point raised by the referee. The performance improvement of U-STN and U-NET does not come from a difference in the number of parameters. While one can easily check the number of parameters within both the architectures using the

`model.summary()` API provided inside the codes, we would like to point out that both these architectures have separately underwent hyperparameter optimization where a much larger network for both U-NET and U-STN has been tested with even more number of filters . The architectures chosen in the manuscript are the most optimal ones based on significant trial and error. Both have roughly the same number of training parameters and it is important to note that U-NET with any larger number of parameters would perform poorly as compared to the reported performance. This is also true for U-STN. In both the architectures, the bottleneck layer has the same size of $8 \times 16$. We have reported this in the caption of Figure 1.

**Referee's comment:**

**1.2 As a result, U-STN may not be physically constrained**
[L43-45] U-STN is specifically introduced as a method to physically constrain a deep learning framework. Even if the authors use nuanced language, I find this motivation misleading for the reader, especially as both examples mentioned by the authors enforce invariance (which is different from equivariance, and even more different from the setup introduced in this manuscript): [2] weakly enforces invariants associated to the Navier-Stokes equations via the loss function, while [1] enforces the PDE structure (associated to invariants) and initial conditions as soft constraints and the boundary conditions as hard constraints in the case of the shallow water equations on a sphere. This is fundamentally different from the more data-driven and flexible approach adopted by the authors in this manuscript: Arguably, no physical constraints are enforced on the outputs of U-STN. Would it be possible to revise the introduction to clarify that this framework is not directly analogous to standard physics-informed approaches that can be found in the literature? Note that these revisions would not necessarily decrease the manuscript's impact but simply clarify the exact motivation (e.g., interpretability) and benefits (e.g., improved accuracy) of this novel architecture.

*Authors' response:* We agree with the referee's point here. The current set-up in this manuscript is very different from physics-informed architectures that enforces the exact PDE and boundary and initial conditions governing the system's dynamics. We thank the referee for this excellent suggestion. We have clarified this in the introduction of the revised manuscript and marked it with blue in Lines 75-76.

**Referee's comment:**

**1.3 However, the current version of U-STN can help gain physical insight**

While U-STN does not seem to enforce physical constraints, its strength could rely on the low-dimensional, interpretable construction of T. Once the authors confirm the superiority of U-STN over U-Net (1.1.2), it would be interesting to better understand how the transformation $T_\theta$ associated to each sample improves the accuracy of the data-driven forecast. This could be done quickly and within the manuscript's scope by uniquely decomposing each transformation T into its corresponding scaling, rotation, and translation (e.g., using this decomposition https://stackoverflow.com/questions/45159314/decompose-2d-transformation-matrix, or any

well-defined decomposition that the authors find interpretable). Once the transformation is decomposed, the authors could answer questions such as:

- How does the transformation vary from sample to sample (e.g., is the scaling, the rotation, or both different)?
- How does the transformation vary from field to field (e.g., does the transformation significantly change when adding temperature as an output)?
- What does this transformation mean physically (e.g., can the rotation or scaling be traced back to atmospheric wave properties)?
- Why does this transformation improve the accuracy of the data-driven forecast (e.g., are these just tunable parameters or is this transformation preventing erroneous assumptions about the rotational symmetry of e.g. wave breaking events as mentioned by the authors)?

This could be a good way to motivate the introduction of the spatial transformer module if it cannot be clearly proven that it preserves equivariance.

*Authors' response:* We appreciate the referee's comments on the interpretation of the transformation matrix $T(\theta)$. As we had indicated in our previous responses, we had tried to perform the decomposition of $T(\theta)$ for different samples as well as different fields following the referee's suggestion. While the parameters $\theta$ are different for different samples and different fields (Z500 and T850), it is hard to interpret and map the nature of transformation with respect to the change in the instantaneous fields especially since the transformation is performed on the low-dimensional space. We have carefully revised the manuscript and ensured that we do not make any claims about $T(\theta)$ capturing the correct representation of wavebreaking events in Lines 258-261 or that it captures any meaningful or interpretable transformations in Lines 264-267. The only evidence that we see in using $T(\theta)$ is an overall improvement in prediction performance.

We would also like to point out that in some of our ongoing work we have developed a more robust framework to perform physically meaningful interpretations of deep networks by using their spectral representations in Fourier space and can attribute the performance improvement of one architecture over another through the Fourier spectrum of the latent space. We intend to extend our framework to interpret the U-STN architecture as well and leave it for future work. We emphasize that the current set-up in this manuscript is difficult to interpret physically as mentioned in Lines 264-267 in the revised manuscript.

**Referee's comment:**

**2 Minor Comments**
[Table 1] (Very minor) Would "change appropriately in response to a transformation" be clearer than "change appropriately to a transformation" here?

*Authors' response:* Thank you for the suggestion. The revised manuscript has been updated.

[Figure 1] Would it be possible to specify the bottleneck's size on this schematic, i.e. the shape of the localization network output?

*Authors' response:* Thank you for the suggestion. While adding the information about the bottleneck size in Figure 1 makes it too busy, we have added the size of the bottleneck layer in the caption of the figure. We have also added the sizes of the outputs after each transformation in the architecture in Table 2.

[L198] Consider replacing R (\cal R) with R (\mathbb{R}) to follow conventions.

*Authors' response:* Thank you for the suggestion. The revised manuscript has been updated with this change.

[Equation 6] This looks like an auto-correlation matrix rather than a covariance matrix: Is this equation correct?

*Authors' response:* This is the equation that is used to calculate the background covariance matrix of the state $Z(t + 24\Delta t)$. A lagged autocorrelation matrix would have the two entries of the inner product differ in the "*lag*" value.

[L272] "not shown for brevity": As the manuscript is already rich in ideas and technical details, I understand the authors' motivation to only focus on one field in the main text. However, since the manuscript significantly gains in generality (from a meteorological perspective) by showing the improved prediction

*Authors' response:* We thank the referee for this suggestion. As we have shown in the previous responses (and in Figure 1 in this response), we have good prediction skill on T850 as well. We have added the figure comparing forecasting performance of T850 between U-STN12 and U-NET12 in Figure 10 in the revised manuscript in the appendix. However, in this manuscript, in order to integrate DA with DDWP and to introduce the multi-step DA framework we have taken Z500 as an example. As we have mentioned in the manuscript as well, we can use any other meteorological variable for this exercise.

[Figure]

Figure 1. Anomaly correlation coefficient (ACC) calculated between T850 anomalies of ERA5 and T850 anomalies predicted using USTN12 and U-NET12 from 30 noise-free, random initial conditions. The solid lines and the shadings show the mean and the standard deviation over the 30 initial conditions.

[Figures 5,6,8,9] Missing units for RMSE (it should be meters if I am not mistaken).

*Authors' response:* We have mentioned the units of RMSE in the caption of each of the figures. Thank you for pointing this out.

[Caption of Figure 5] The authors use parentheses for clarification, abbreviations, references, and I find that using it to express opposites is confusing in this caption (see [4] for a general discussion on the topic). More specifically, it looks like the coefficient of determination R is used to abbreviate RMSE. Would it be possible to rewrite the caption to avoid potentially confusing the reader?

*Authors' response:* Thank you. We have re-written the caption.

[Figure 6, L275-280] Since it is difficult for readers to visualize numbers that are written in text, I highly recommend adding the WeatherBench baselines (and maybe [5, 3]) as scattered crosses on

the left plot of Figure 6. This would facilitate the comparison between the performance of these baselines and U-STN, which would further underline the good performance of U-STN.

*Authors' response:* We thank the referee for this suggestion. We had provided the comparison on RMSE in our previous response as well and we further provide that table in this response (Table 1). We have now added an appendix in the revised manuscript between Lines 390-391, where we have provided the comparison of RMSE between U-STN12 and two WeatherBench models in Table 3.

However, we do not intend this paper to be a comparison between different DDWP models on WeatherBench and other papers. Currently, WeatherBench has a leaderboard where we intend to submit our model for evaluation and comparison. But in this paper, we want to introduce DA to be integrated with DDWP and the novel multi-step DA framework.

**Table 1.** A comparison between U-STN12 presented in this manuscript and the linear regression and CNN models from WeatherBench [1]. Here, we have used the RMSE values of direct linear regression and CNN prediction at day 3 and day 5. Here "direct" refers to the models trained to predict Z500 directly at 3 and 5 days instead of iteratively predicting every hour.

| Models | RMSE $(m^2 s^{-2})$ at 3 days | RMSE $(m^2 s^{-2})$ at 5 days |
|---|---|---|
| Linear regression (direct) from WeatherBench | 693 | 783 |
| CNN (direct) from WeatherBench | 626 | 757 |
| U-STN12 (our model) | **294** | **490** |

[L320] Typo: "At 24th hours".

*Authors' response:* Thank you for pointing this out. We have now fixed it in the revised manuscript.

[L382] "forecast/assimilation": Do the authors mean "forecast and assimilation"? The current phrasing may confuse readers.

*Authors' response:* Yes, that is what we meant. We have now revised this line in the revised manuscript.

[Code availability statement] To clarify, I was referring to the authors' GitHub repository (and not the WeatherBench repository). Would it be possible to share the GitHub of this manuscript specifically (corresponding to the Zenodo URL https://zenodo.org/record/5553570#.YX-QKp5KhPY) as part of the code availability statement?

*Authors' response:* We thank the referee for this excellent suggestion. However, it seems that GMD does not allow us to put a Github link for our own code in the code availability statement and as soon as we submit, we receive our manuscript back for a revision that indicates this issue

with Github links which are not persistent archives. However, we intend to put the Github link of our codes on the arxiv pre-print. For the referee's ease of accessibility, we provide the Github link for our codes herein: https://github.com/ashesh6810/DDWP-DA.

**References**

1. Jaderberg, M., Simonyan, K., Zisserman, A., et al.: Spatial transformer networks, in: Advances in Neural Information Processing Systems, pp. 2017–2025, 2015.

---

## Author Response (AR4)

*Referee's comment:*

*Thank you again for thoroughly responding to the reviewers' comments: At this point, the manuscript is clear and the authors' framework reproducible. However, the manuscript's first key point remains to be proven because (1) UNETx and U-STNx are not compared in a systematic fashion (1.1); and (2) U-STN is still misleadingly presented as equivariance-preserving in parts of the manuscript (1.2). Additionally, WeatherBench models may be unfairly compared to a version of U-STN including data assimilation cycles (1.3), and U-STN is still framed as a physics-informed machine learning (ML) framework (1.4). Finally, there may still be discrepancies between the shared code and what is reported in the manuscript (2.1). I continue to think that this manuscript will be a welcome contribution to GMD once revised so as to not mislead readers about equivariance preservation. Overall, it would be helpful to clarify whether the improved test performance of U-STNx reported by the authors without data assimilation (e.g., Fig 6) is truly the result of some form of equivariance preservation, or whether using a bottleneck and a spatial transformer simply prevents overfitting the WeatherBench training set.*

**Authors' response:**

We thank the referee for their insightful comments which have substantially improved the quality and accessibility of the manuscript. Herein, we address each of the referee's comments. All the changes in the revised manuscript have been tracked in blue (note that the tracked manuscript tracks all changes from the original submission) until round 2 and the last round of revision has been tracked in red.

*We want to emphasize upfront regarding the major issue that the referee has been pointing out about the equivariance-preserving nature of the network:* we agree with the referee that having an equivariance-preserving latent space via a spatial transformer is not the same as a fully equivariant network. Hence, we have decided to remove *"equivariance-preserving"* throughout the manuscript. Instead, whenever we have talked about the advantages of the spatial transformer, we have explained that it *may help* in capturing rotational and scaling features only in the transformation in the latent space where it has been implemented. We hope that this would avoid any confusion amongst readers that our network is indeed *not fully equivariant end-to-end*.

*Referee's comment:*

**1.1 Comparison between U-NET and U-STN**
*The first key point of the manuscript relies on comparing U-STN with U-NET to establish U-STN's improved performance compared to U-NET ("improving the forecast skill by 45%"). This requires being as precise as possible when comparing*
*U-STN and U-NET:*
- *According to Table 2, U-STN has three additional layers (10/11/12) of 200/100/50 neurons. Would it be possible to transparently list the number of trainable parameters (e.g., the output of model.summary()) for both the U-STN and U-NET? If they are different, what would suggest that this is a fair comparison?*

- *For example, how is it possible to know whether U-STN improved performance over the test set comes from the spatial transformer or from the bottleneck created by layers 10/11/12. Would it be possible to introduce a U-NET with the exact same architecture as U-STN (minus the spatial transformer) to test whether the improved performance simply comes from not overfitting the WeatherBench training set?*

- *"the optimal set of hyperparameters that have been obtained after extensive trial and error": This is a vague statement in the age of formal hyperparameter optimization (e.g., Hyperas, Hyperopt, the Keras tuner, SHERPA, etc.) that does not establish in any way that the architecture of U-STN is comparable to that of U-NET. Would it be possible to:*
  *- List the hyperparameters and their considered range during the manual search?*
  *- Systematically report the performance of U-NET and U-STN over the training/validation/test set (as opposed to just the test set) to identify whether U-STN is a better fit, whether it generalizes better, etc.?*
  *- This could also help better understand the role of the spatial transformer (e.g., does it help because it prevents overfitting the training set?) as equivariance is not generally preserved by the U-STN framework.*

**Authors' response:**
We thank the referee for this insightful comment about the relative complexity of U-STN and U-NET and how that affects the prediction horizon of each of these models. We would first like to point out that the U-STN and U-NET have **separately** undergone hyperparameter optimization (albeit via trial and error, nonetheless, an extensive one). This is to say, that a U-NET with more depth, more width (large number of filters), or larger/smaller kernel sizes would not have better prediction horizon than the current U-NET. So, comparing the number of parameters in U-STN and U-NET separately does not lead to any conclusion about their relative performance. Still, based on the referee's comment, we have added the model.summary() command in our codes to transparently report the total number of trainable parameters in each of the models (U-STN: 2,461,965, U-NET: 291,777). Of course, this discrepancy in the number of parameters is largely due to the dense layers in the U-STN. Further, herein, we include a few of the tables that we have obtained from our hyperparameter tuning experiments to show that larger number of parameters in the U-NET would not lead to better prediction horizons in Table 1 of the paper.

- A larger number of parameters in the U-NET as compared to the U-STN would not lead to a better model (longer prediction horizon) since the depth, width, and kernel sizes have been optimized for this U-NET through hyperparameter tuning. It must be noted, that owing to a limited size of the training set, an overly complex model may not necessarily lead to better performance. We report some of the results on hyperparameter tuning in Table 1.

- We have performed experiments with adding different number of dense layers to the U-NET when performing hyperparameter tuning. We show the prediction horizons on the validation set in Table 2. As is evident from Table 2, adding these dense layers in the U-

NET does not lead to any improvement and in most cases deteriorates the performance of the U-NET.

Table 1, and Table 2 clearly shows that simply increasing the number of parameters in the U-NET would not necessarily lead to better prediction horizons. While we have done further hyperparameter tuning, these two tables summarize that neither including the dense layers (as in U-STN) or increasing the number of filters lead to better prediction horizon. Based on significant trial and error, we believe that U-STN outperforms an optimal U-NET in this problem. We hope that the referee finds these hyperparameter tuning logs sufficient to show that the spatial transformer layer does have an added advantage as compared to regular U-NET.

Table 1. Hyperparameter tuning for number of filters in each layer and kernel sizes on the prediction horizon of U-NET on the validation dataset. Note that increasing number of filters increases the number of trainable parameters. **Note that the last row of this table (red) has 13,773,665 trainable parameters which is more than that of U-STN but does not show improvements in performance.** The highlighted element in the table is the optimal number of filters in each layer with the optimal kernel size.

| Number of Filters | Kernel Sizes | | | | |
|---|---|---|---|---|---|
| | **3x3** | **5x5** | **7x7** | **11x11** | |
| **16** | ~77 | ~77 | ~75 | ~75 | Prediction Horizon (hrs) |
| **32** | ~88 | ~96 | ~93.3 | ~91 | |
| **64** | ~82 | ~95.3 | ~92 | ~81 | |
| **128** | ~88 | ~95 | ~88 | ~88 | |
| **256** | ~74 | ~74 | ~72.2 | ~68 | |

Table 2. Hyperparameter tuning for number of filters and dense layers before the bottleneck on the prediction horizon of U-NET on the validation dataset.  Here as well, we can see that the optimal performance of the U-NET can be obtained without any dense layers in between. Note that the last row of this table has two orders of magnitude more parameters than the optimal U-STN.

| Number of Filters | No dense layers | 3 Dense layers, 200-100-50 | 4 Dense layers, 300-200-100-50 | 5 Dense layers, 400-300-200-100-50 | |
|---|---|---|---|---|---|
| **16** | ~77 | ~62 | ~62 | ~61.5 | Prediction Horizon (hrs) |
| **32** | ~96 | ~78 | ~72 | ~67 | |
| **64** | ~95.3 | ~72 | ~69 | ~67 | |
| **128** | ~95 | ~72 | ~68 | ~68 | |
| **256** | ~74 | ~72 | ~68 | ~66 | |

*Referee's comment:*

*1.2 U-STN is not equivariance-preserving*
*U-STN is still framed as a framework preserving equivariance:*
*[L4 in the abstract] "to preserve a property called equivariance"*
*[L44] "based on building physical properties called equivariance"*
*[L102] "Introducing the equivariance-preserving"*
*[L338] "to preserve equivariances".*
*While the authors added a discussion at the end of subsection 3.1.2, the above statements would mislead most readers (especially those focusing on the abstract/introduction/conclusion) into thinking that this framework ensures equivariance preservation. Assuming that the above claims relate to SO (3) equivariance, which if I am not mistaken would here be defined along the lines of:*

$$\forall\, \theta, \forall Inputs, U - STN[T_\theta(inputs) = \tau_\theta(U - STN(inputs))$$

*would it be possible to clarify that:*
- *The framework is not equivariant because it learns one set of parameters per sample, and that these parameters are not physically interpretable according to the authors (in their response to reviews),*
- *The U-STN architecture includes skip-connections, which prevent equivariance preservation at multiple levels of the architecture (up6 and up7 in the code use conv2 and conv1).*

*Note that other manuscripts using equivariance preservation (e.g., [1]) typically define equivariance mathematically and exactly stipulate whether the property is satisfied or not upfront.*

**Authors' response:**

We agree with the referee that using the term, *"equivariance-preserving"* can be misleading to the audience. Hence, we have decided to remove this term whenever referencing our network within the revised manuscript. Instead, whenever we talk about the advantages of the spatial transformer, we would explain that *it may help in capturing rotational and stretching features* in the transformation of the latent space. We have further added Lines 153-154 explicitly mentioning that the entire network is not equivariant by construction. We hope that this change would remove all confusion regarding the network's properties amongst the readers. We thank the referee for pointing this out and having the invigorating discussion.

*Referee's comment:*

*1.3 Misleading comparison with WeatherBench*
*[Table3, Sec 6.2]*
*1. Does the U-STN12 compared to WeatherBench models assimilate data from the test set? In the affirmative, I*
*recommend comparing WeatherBench models to the version of U-STN12 that does not use data assimilation (e.g., the version presented in Fig 6) and clarifying the text accordingly.*

*2. Are the authors using the same test set (their paper uses 2018) as the test set used to report the WeatherBench models' performance (2017 and 2018 according to [2])? This would also prevent a fair comparison and I recommend using the same test set as in WeatherBench while explicitly acknowledging that samples from the validation set will be used for this particular comparison.*

**Authors' response:**
We thank the referee for raising this point. When comparing with WeatherBench we have used U-STN12 **without DA.** We have indicated that in the title of section 4.1 as well as in the appendix in section 6.2 (Line 390).

We further want to emphasize that the prediction of U-STN12 or U-NET12 will not change with the choice of the year in the testing dataset (we have tested on data from 2017, 2016, and 2015 as well; for these years the training samples would be reduced as well). We have reported prediction horizons across multiple initial conditions sampled from the testing dataset.

Based on the referee's suggestion, we include Table 3 showing the prediction horizon over 30 initial conditions, sampled from years 2017, 2016, and 2015.

Table 3. Prediction horizon of U-STN12 and U-NET12 for testing years 2017, 2016, and 2015.

| Model | 2017 | 2016 | 2015 | |
|---|---|---|---|---|
| U-STN12 | $120 \pm 6$ | $120 \pm 4$ | $120 \pm 3$ | Prediction |
| U-NET12 | $96 \pm 8$ | $96 \pm 6$ | $96 \pm 6$ | Horizon (hrs) |

*Referee's comment:*

*1.4 U-STN is still framed as a physics-informed ML framework*
*U-STN is still framed as a framework to improve physical consistency:*
*[Title] Towards physically-consistent [L3 in the abstract] improve their physical consistency*
*[L44] provide a framework for [L41-42] incorporating physical constraints into the often physics-agnostic ML models*
*However, given that U-STN does not preserve equivariance (see 1.2), it is unclear why U-STN would make the predictions more physically consistent (other than the qualitative description in Sec 4.1). If it is impossible to quantitatively establish the improved physical consistency of U-STN predictions, I suggest removing the claims listed above.*

**Authors' response:**
Thank you. Based on the referee's suggestion, we have removed these claims. We have changed the word "physical consistency" through-out the manuscript. However, it must be noted that capturing rotation and scaling (albeit in the encoding of the latent space) is motivated from physics. Therefore, we use the word, "physics-inspired". In Lines 41-42 (revised manuscript), we have talked about incorporating physical constraints as a general procedure to improve DDWP models and have not intended to be a description of our DDWP model. We have further clarified in the Lines 75-76 that our DDWP model is not a traditional physics-informed neural network.

*2 Minor Comments*
*2.1 Possible discrepancies between the code and the manuscript*
*In the accompanying code for the baseline U-NETx, it seems that the authors are still training a U-STN ("stn()") for most of the training files instead of a U-NET ("unet_baseline()"). Would it be possible to explain this apparent discrepancy,*
*which appears in all of the U-NETx (x = 1hr; 12hr, etc.) scripts?*

**Authors' response:**
Thanks for this. We had previously defined the baseline model as stn() as well, just for ease of imports. We have now changed the name of the functions to reflect baseline and U-STN separately. We have accordingly updated both the Github and the Zenodo links.

*2.2 Typos*
*[L26-27] "e.g." should be moved to the beginning of the parentheses*
*[L131] "indicates the  Δt that is used": Should e.g., "in units hours" be added to specify the units of Δt in the U-STNx acronym?*
*[L352] "data-drivenly": Is this grammatically correct or should "in a data-driven fashion" be used instead here?*
*[L387] "have shown" -> "show"*

**Authors' response:**
Thanks for pointing these out. We have now fixed them based on the referee's suggestions.